# CLIP-PAE: Projection-Augmentation Embedding to Extract Relevant Features for a Disentangled, Interpretable, and Controllable Text-Guided Image Manipulation

## Abstract

Recently introduced Contrastive Language-Image Pre-Training (CLIP) (Radford et al., 2021) bridges images and text by embedding them into a joint latent space. This opens the door to ample literature that aims to manipulate an input image by providing a textual explanation. However, due to the discrepancy between image and text embeddings in the joint space, using text embeddings as the optimization target often introduces undesired artifacts in the resulting images. Disentanglement, interpretability, and controllability are also hard to guarantee for manipulation. To alleviate these problems, we propose to define corpus subspaces spanned by relevant prompts to capture specific image characteristics. We introduce CLIP *projection-augmentation embedding* (PAE) as an optimization target to improve the performance of text-guided image manipulation. Our method is a simple and general paradigm that can be easily computed and adapted, and smoothly incorporated into *any* CLIP-based image manipulation algorithm. To demonstrate the effectiveness of our method, we conduct several theoretical and empirical studies. As a case study, we utilize the method for text-guided semantic face editing. We quantitatively and qualitatively demonstrate that PAE facilitates a more disentangled, interpretable, and controllable image manipulation with state-of-the-art quality and accuracy.

## 1 Introduction

In text-guided image manipulation, the system receives an image and a text prompt and is tasked with editing the image according to the text prompt. Such tasks have received high research attention due to the great expressive power of natural language. Recently introduced and increasingly popular Contrastive Language-Image Pre-Training (CLIP) (Radford et al., 2021) is a technique to achieve this by embedding images and texts into a joint latent space. Combined with generative techniques such as Generative Adversarial Networks (GANs) (Goodfellow et al., 2014) and Diffusion (Ho et al., 2020; Dhariwal & Nichol, 2021), CLIP has been utilized to develop several high-quality image manipulation methods (*e.g.*, Ling et al., 2021; Zhang et al., 2021; Antal & Bodó, 2021; Khodadadeh et al., 2022), where the image is optimized to be similar to the text prompt in the CLIP joint space.

There are three important but difficult to satisfy properties when performing image manipulation: disentanglement, interpretability, and controllability. Disentanglement means that the manipulation should only change the features referred to by the text prompt and should not affect other irrelevant attributes (Wu et al., 2021; Xu et al., 2022). Interpretability means that we know why/how an edit to the latent code affects the output image and thus we understand the reasoning behind each model decision (Doshi-Velez & Kim, 2018; Miller, 2019), or that the model can extract relevant information from the given data (Murdoch et al., 2019). Finally, controllability is the ability to control the intensity of the edit (You et al., 2021; Park et al., 2020; Li et al., 2019) for individual factors and hence is tightly related to disentanglement.

In CLIP-based text-guided image manipulation methods, since both the latent space of the generative network and the embedding space of CLIP extensively compress information (*e.g.*, an $1024 \times 1024$

image contains $3 \times 1024^2$ dimensions, whereas a typical StyleGAN (Karras et al., 2019) and CLIP both only have 512-dimensional latent spaces), the manipulation in the latent space is akin to a black box. Gabbay et al. (2021) argue that a GAN fails at disentanglement because it only focuses on localized features. Zindancıoğlu & Sezgin (2021) also show that several action units in StyleGAN (Karras et al., 2019) are correlated even if they are responsible for semantically distinct attributes in the image. In addition, we found that although CLIP embeddings of images and texts share the same space, they actually reside far away from each other (see Section 3.1), which can lead to undesired artifacts in the generated images such as unintended editing or distortion of facial identity (see Figure 1). Finally, most existing methods for CLIP-based text-guided image manipulation do not allow for free control of the magnitude of the edit. As a result, how to perform image manipulation in a disentangled, interpretable, and controllable way with the help of text models remains a hard and open problem.

In this paper, we introduce a technique that can be applied to *any* CLIP-based text-guided image manipulation method to yield a more disentangled, interpretable, and controllable performance. Rather than optimizing an image directly towards the text prompt in the CLIP joint latent space, we propose a novel type of CLIP embedding, *projection-augmentation embedding* (PAE), as the optimization target. PAE was motivated by two empirical observations. First, the images do not overlap with their corresponding texts in the joint space. This means that a text embedding does not represent the embedding of the true target image that should be optimized for. Therefore, directly optimizing an image to be similar to the text in the CLIP space results in undesirable artifacts or changes in irrelevant attributes. Second, a CLIP subspace constructed via embeddings of a set of relevant texts can constrain the changes of an image with only relevant attributes.

To construct a PAE, we first project the embedding of the input image onto a corpus subspace constructed by relevant texts describing the attributes we aim to disentangle, and record a residual vector. Next, we augment this projected embedding in the subspace with the target text prompt, allowing for a user-specified *augmenting power* to control the intensity of change. Finally, we add back the residual to reconstruct a vector close to the "image region" of the joint space. We demonstrate that the PAE is a better approximation to the embedding of the true target image. With PAE, we achieve better interpretability via an explicit construction of the corpus subspace. We achieve better disentanglement since the subspace constrains the changes of the original image with only relevant attributes. We achieve free control of the magnitude of the edit via a user-specified augmenting power. PAE is easy to use, quick to pre-compute, and can be incorporated into *any* CLIP-based latent manipulation algorithms to improve performance.

In short, we highlight the three major contributions of this paper:

1. (Section 3) We perform a series of empirical analyses of the CLIP space and its subspaces, identifying i) the limitations of using a naive CLIP loss for text-guided image editing; and ii) several unique properties of the CLIP subspace.

2. (Section 4) Based on our findings in Section 3, we propose the *projection-augmentation embedding* (PAE) as a better approximation to the embedding of the true target image.

3. (Section 5) We demonstrate that employing PAE as an alternative optimization target facilitates a more disentangled, interpretable, and controllable text-guided image manipulation. This is validated through several text-guided semantic face editing experiments where we integrate PAE into a set of state-of-the-art models. We quantitatively and qualitatively demonstrate that PAE boosts the performance of all chosen models with high quality and accuracy.

## 2    RELATED WORK

**Latent Manipulation for Image Editing**    One popular approach for image manipulation is based on its latent code: the input image is first embedded into the latent space of a pre-trained generative network such as GAN (Goodfellow et al., 2014), and then to modify the image, one updates either the latent code of the image, (*e.g.*, Zhu et al., 2016; Ling et al., 2021; Zhang et al., 2021; Antal & Bodó, 2021; Khodadadeh et al., 2022; Creswell & Bharath, 2018; Perarnau et al., 2016; Pidhorskyi et al., 2020; Hou et al., 2022b; Xia et al., 2021; Kocasari et al., 2022; Patashnik et al., 2021; Shen et al., 2020), or the weights of the network (*e.g.*, Cherepkov et al., 2021; Nitzan et al., 2022; Reddy

et al., 2021) to obtain the desired image editing. However, these methods can only alter a set of pre-defined attributes and thus lack flexibility and generalizability.

**CLIP for Text-Guided Image Manipulation**  In 2021, Radford et al. proposed Contrastive Language-Image Pre-Training (CLIP), where an image encoder and a text encoder are trained such that the semantically similar images and texts are also similar in the joint embedding space. The insight of connecting image and text in the same space brings up a wide spectrum of applications in computer vision tasks. For example, Radford et al. provide applications for image captioning, image class prediction, and zero-shot transfer. More sophisticated tasks include language-driven image generation (Ramesh et al., 2022), zero-shot semantic segmentation (Li et al., 2022), image emotion classification (IEC) (Deng et al., 2022), large-scale detection of specific content (González-Pizarro & Zannettou, 2022), object proposal generation (Shi et al., 2022b), object sketching (Vinker et al., 2022) and referring expression grounding (Shi et al., 2022a).

CLIP has also been applied to text-guided image manipulation tasks. In this domain, one approach is to edit the latent code of a certain generative network so that the embedding of the generated image is similar to the embedding of the given text in the CLIP space (Kocasari et al., 2022; Patashnik et al., 2021; Xia et al., 2021; Hou et al., 2022a; Ramesh et al., 2022) (see Figure 4a). However, this straightforward approach sometimes fails to change the desired attributes, or fails to change them in a disentangled way: other unrelated features are also affected (see the comparative study in Sections 5.2 and 5.3). In addition, the proposed methods often fail to exhibit enough interpretability and controllability. This is probably attributed to the separation of images and texts in the CLIP embeddings space, so that optimizing an image embedding towards a text embedding naturally introduces some undesired effects. Although there are some remedies, such as penalizing the change of features in the editing (*e.g.*, (Canfes et al., 2022; Kocasari et al., 2022)), separating the image into different granule levels (Patashnik et al., 2021), formulating a constrained optimization problem (Zhu et al., 2016), partially labeling a set of features (Gabbay et al., 2021), or updating the parameters of the underlying GAN to preserve features (*e.g.*, (Nitzan et al., 2022; Reddy et al., 2021; Cherepkov et al., 2021)), they do not collectively achieve a disentangled, interpretable, and controllable manipulation. Additionally, some of them target all features indifferently: they still do not separate the related features from the irrelevant ones. Moreover, most of these techniques only work for specific methods or attributes and lack generalizability, and some of them are too complicated and time-consuming, hence cannot be used in practical large-scale applications.

## 3    ANALYSIS OF THE CLIP JOINT SPACE AND SUBSPACE

Most CLIP-based text-guided image manipulation algorithms (*e.g.*, Kocasari et al., 2022; Patashnik et al., 2021; Xia et al., 2021; Hou et al., 2022a; Ramesh et al., 2022) follow a general paradigm where certain parameters of an image editing process (such as the latent code or the weights of a generative network) are trained to minimize a *cosine similarity loss* between the resulting image and a text prompt in the CLIP joint space (see Figure 4a). However, naively using this loss may introduce undesirable artifacts or unintended changes unrelated to the text prompt, as shown in Figure 1. We hypothesize that this is due to the discrepancy between the images and the texts in the CLIP joint space, as we will demonstrate in Section 3.1. Therefore, the embedding of the text prompt in the CLIP joint space does not actually represent the embedding of the true target image that should be optimized for. To alleviate this issue, we construct CLIP subspaces with desirable properties (see Section 3.2) leading to our proposed projection-augmentation embedding (see Section 4).

### 3.1    NON-OVERLAPPING IMAGE AND TEXT EMBEDDINGS

As an empirical demonstration, we collect over 1500 face images and 1500 textual descriptions of faces (*e.g.*, emotion, hairstyle or general) and visualize their embeddings in the CLIP joint space using PCA (Jolliffe, 2002) in Figure 2. Note that the visualization is given using Euclidean distance; Although CLIP is trained with cosine similarity, if we normalize all the embeddings, the Euclidean distance and cosine distance give exactly the same ranking because

$$\|\boldsymbol{a} - \boldsymbol{b}\|_2 = \sqrt{(\boldsymbol{a} - \boldsymbol{b})^2} = \sqrt{\boldsymbol{a}^2 + \boldsymbol{b}^2 - 2\boldsymbol{a}^T\boldsymbol{b}} = \sqrt{2 - 2\mathrm{CosSim}(\boldsymbol{a}, \boldsymbol{b})}. \qquad (1)$$

As shown in Figure 2, the image and the text embeddings lie in two non-overlapping regions. They also exhibit a lower inter-modality cosine similarity compared to the intra-modality similarity, re-

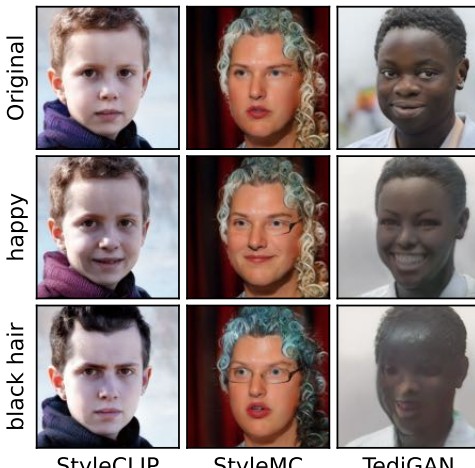

Figure 1: Using text embedding as the optimization target results in unsatisfactory outputs. Note the entangled changes (*e.g.*, clothing of the left child, glasses of the lady), inaccurate output, and artifacts.

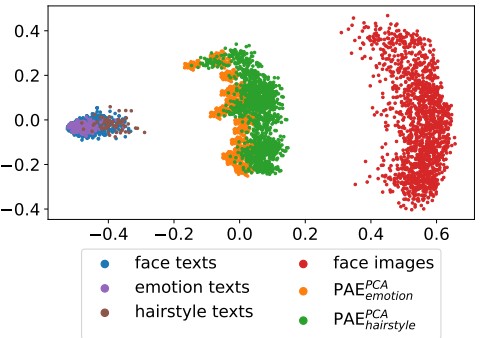

Figure 2: PCA visualization of face images and text descriptions (Section 3.1), and their corresponding PAEs in the CLIP space (Section 4). Note that image and text embeddings are non-overlapping, that PAEs are closer to image, and that PAEs spread less widely than the face images, indicating that they contain more specified information.

gardless of their semantic meanings. For example, the cosine similarity between an image of a dog and the text "dog" is 0.253 and that between an image of a cat and the text "cat" is 0.275. On the other hand, the similarity between a dog image and a cat image reaches 0.841, and that between the two texts "dot" and "cat" is as high as 0.936, much higher than the intra-modality similarity. We record additional evaluations of inter-modality and intra-modality cosine similarities in Appendix A.1. In the general paradigm described above (and see Figure 4a), the CLIP embedding of the ideal target image is essentially approximated by the embedding of the text description. However, the separation of text and image embeddings clearly invalidates such approximation and thus leads to artifacts (see Figure 1 and more in Sections 5.2 and 5.3).

## 3.2 Subspaces distilling relevant information from image embeddings

Since the joint space is a vector space, we can construct a subspace of it using a set of relevant text prompts as basis vectors. For example, we can construct an "emotion subspace" using relevant emotion texts. In this section, we explain how such a subspace distills relevant information from images, which can be used to constrain the changes of an image. We also include some experiments in Appendix A.5 to show that the subspace can distill information from texts. We leave the mathematical details of the formulation of the subspace to Section 4, but preview certain properties of the subspace as these properties inspire the formulation of our method.

We invited ten participants to record five-second videos of their face changing from neutral to laughing out loud (LOL). We compute the cosine similarity (averaged over all videos) of each frame to the first frame or to the text "a happy face" in the CLIP space, or after projecting them onto the emotion or hairstyle subspace. The results are shown in Figure 3.

In Figure 3a, the similarity decreases much faster in the emotion subspace compared to the CLIP space and the hairstyle subspace. A similar pattern shows up in Figure 3b — the similarity increases much faster in the emotion subspace. Since the subspace has a lower dimensionality than the original space, these observations indicate that the information regarding the irrelevant attributes is *discarded* during the projection. Hence the changes of the hairstyle in the facial images have a low effect influencing the similarity — the emotion subspace only *distills* emotional attributes from the image and discards others. This inspires the formulation of our method: if we make changes to the embedding of the original image *within a subspace*, it would only induce a change of attributes related to the subspace. As a result, these changes are *disentangled*.

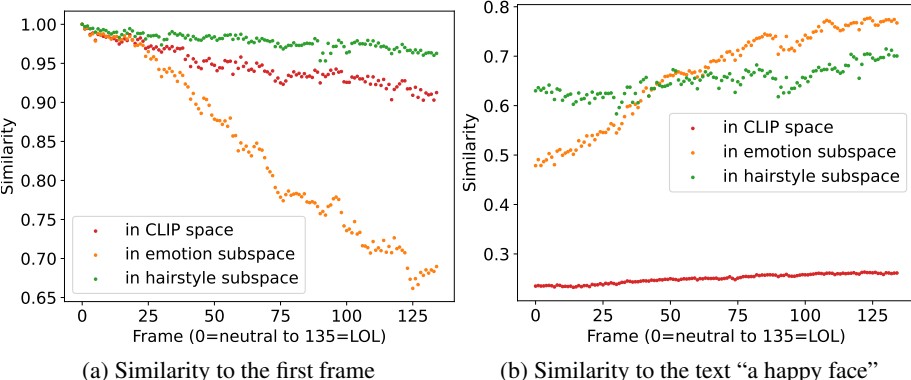

(a) Similarity to the first frame

(b) Similarity to the text "a happy face"

Figure 3: Similarity of video frames to the first frame and to the text "a happy face" in different spaces. The changes in the emotion subspace are the most significant as it distills the relevant information.

## 4 PROJECTION-AUGMENTATION EMBEDDING

Motivated by our findings in Section 3, we propose the *projection-augmentation embedding* (PAE) as a better approximation to the embedding of the true target image. There are three objectives we aim to fulfill when constructing such an embedding: i) it should be closer to the image region than the text is; ii) it should be guided by the target text prompt; iii) the guidance should be provided within the subspace so the changes are disentangled.

### 4.1 OVERVIEW

Given an input image $I$ and a text prompt $T$, we construct the PAE, denoted as $\mathcal{E}_{\mathfrak{W}}(I, T, \alpha)$, as follows. First, we obtain the embeddings of $I$ and $T$, denoted as $\boldsymbol{e}_I$ and $\boldsymbol{e}_T$, via the CLIP image and text encoders (Radford et al., 2021), then we project $\boldsymbol{e}_I$ onto a corpus subspace $\mathfrak{W}$ constructed with texts describing the attributes we aim to disentangle:

$$\boldsymbol{w} = \mathcal{P}_{\mathfrak{W}}(\boldsymbol{e}_I), \tag{2}$$

where $\mathcal{P}_{\mathfrak{W}}$ is the *projection* operation onto the subspace $\mathfrak{W}$. We will explain the details of $\mathcal{P}_{\mathfrak{W}}$ and the construction of $\mathfrak{W}$ in Section 4.2. Next, we record a *residual* vector $\boldsymbol{r}$ of the projection:

$$\boldsymbol{r} = \boldsymbol{e}_I - \boldsymbol{w}. \tag{3}$$

After that, we *augment* the influence of the text prompt $T$ to the projected embedding $\boldsymbol{w}$, and finally, add back the residual vector $\boldsymbol{r}$ to construct the final embedding:

$$\mathcal{E}_{\mathfrak{W}}(I, T, \alpha) = \mathcal{A}_{\mathfrak{W}, T}(\boldsymbol{w}, \alpha) + \boldsymbol{r}, \tag{4}$$

where $\mathcal{A}_{\mathfrak{W}, T}$ is the augmentation operation in $\mathfrak{W}$ according to $T$ with a controllable *augmenting power* $\alpha$, as will be explained in details in Section 4.3.

A graphic illustration of PAE can be found in Figure 4c. In response to our aforementioned three objectives, we highlight that we add back the residual to ensure that the PAE is close to the image region rather than the text region, that we apply the augmentation to ensure that the PAE is guided by the target text prompt, and that we apply the projection to ensure that the guidance is provided via the subspace so the changes of the image are *disentangled* (recall Section 3.2). As we will see in Section 4.2, the construction of the subspace with explicit selections of relevant prompts makes our approach more *interpretable* compared to black-box latent manipulations. Our introduction of the augmenting power (see Section 4.3) makes it possible to *freely control* the magnitude of the change to the image. PAE can be integrated into any CLIP-based text-guided image manipulation algorithms in place of the text embedding in the final loss function, as shown in Figure 4b.

In Figure 2 we also include corresponding PAEs generated by the same face images and facial descriptions in Section 3.1. We see that compared to the text embeddings, PAEs are indeed closer

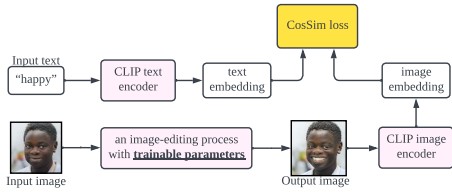

(a) A general paradigm followed by most CLIP-based text-guided image manipulation algorithms for semantic face editing (*e.g.*, Patashnik et al., 2021; Kocasari et al., 2022; Xia et al., 2021).

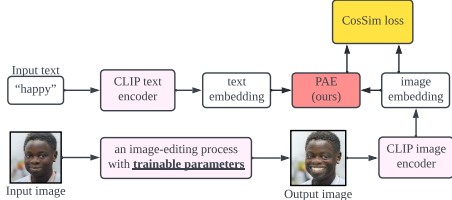

(b) Integration of PAE into the CLIP-based text-guided image manipulation paradigm.

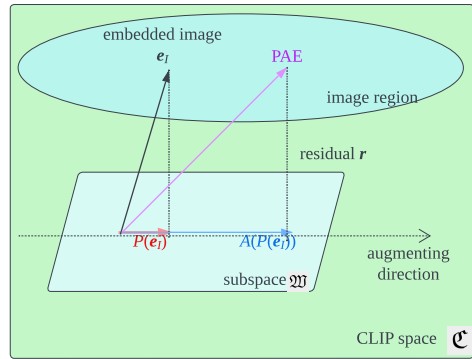

(c) A graphical demonstration of the calculation of PAE. It is calculated by 1. projecting the embedded image $e_I$ (black) to a pre-defined subspace $\mathfrak{W}$ of interest (red); 2. augmenting the projected vector in a way that the effect of text is amplified (blue); and finally 3. adding back the residual $r$ to return to the "image region" in $\mathfrak{C}$ (purple).

Figure 4

to the image embeddings, and that the PAE distribution has a lower variation, suggesting that they contain more specified information.

In the following subsections, we introduce the details of the projection and augmentation operations.

## 4.2 PROJECTION

We introduce two options for the projection operation: $\mathcal{P}_{\mathfrak{W}}^{\text{GS}}$ and $\mathcal{P}_{\mathfrak{W}}^{\text{PCA}}$. $\mathcal{P}_{\mathfrak{W}}^{\text{GS}}$ aims to find the semantic vectors for $\mathfrak{W}$ as the subspace is a semantically meaningful structure. For example, if we aim to change/disentangle the facial emotions, we can use the text embeddings of a set of basic emotions as the basis vectors.

After selecting the basis vectors $\{b_n\}_n$, we apply the **G**ram-**S**chmidt process to obtain an orthonormal basis $\{\hat{b}_n\}_n$, and then project $e_I$ onto $\mathfrak{W}$ by computing a dot product with the basis vectors:

$$\mathcal{P}_{\mathfrak{W}}^{\text{GS}}(e_I) := \sum_k \left( \hat{b}_k^T e_I \right) \hat{b}_k. \tag{5}$$

$\mathcal{P}_{\mathfrak{W}}^{\text{GS}}$ will fail if there is no apparent semantic basis (*e.g.*, for hairstyle, it is hard to find "basic hairstyles"). $\mathcal{P}_{\mathfrak{W}}^{\text{PCA}}$ aims to find the basis of $\mathfrak{W}$ by constructing a set $\mathcal{T}$ consisting of a corpus of relevant text embeddings and performing principal component analysis (PCA) (Jolliffe, 2002) to extract a pre-defined number $N$ of principal components as the basis $\{b_n\}_{n=1}^N$. Effectively, the space spanned by $\{b_n\}_{n=1}^N$ would be a space approximating $\mathfrak{W}$ that encompasses all the related texts in the corpus of interest. After we get the basis, the projection is performed the same way as in Eq. (5). Other dimension reduction techniques can also be used in replace of PCA, such as kernal PCA (Schölkopf et al., 1997) or t-SNE (Van der Maaten & Hinton, 2008)).

We also experimented with a simpler idea of projecting both $e_I$ and $e_T$ onto $\mathfrak{W}$ and perform the optimization in $\mathfrak{W}$, instead of augmenting in $\mathfrak{W}$ and adding back the residual. However, this approach introduced significant artifacts and entangled changes, possibly due to the loss of information during the projection (see Appendix A.4 for more details).

## 4.3 AUGMENTATION

The augmentation operation directs embedding of the original image towards the target. As discussed above, the augmentation on the projected image $w$ should be guided by the target text prompt.

A simple augmentation operation can be

$$\mathcal{A}_{\mathfrak{W},T}(\boldsymbol{w}, \alpha) = \boldsymbol{w} + \alpha \boldsymbol{e}_T. \tag{6}$$

However, in our pilot studies, we found that this simple operation results in a PAE too similar to the original embedding, leading to an optimization process barely doing anything (see Appendix A.3). We strengthen the influence of $\boldsymbol{e}_T$ on $\boldsymbol{w}$ by weakening the unintended attributes of $\boldsymbol{w}$ — components of $\boldsymbol{w}$ where $\mathcal{P}_{\mathfrak{W}}(\boldsymbol{e}_T)$ has a low value — while preserving the sum of coefficients of $\boldsymbol{w}$.

Mathematically, we first calculate the coefficients $c_k$ of $\boldsymbol{w}$ and the coefficients $d_k$ of the projected text $\mathcal{P}_{\mathfrak{W}}(\boldsymbol{e}_T)$, expressed under the basis $\{\boldsymbol{b}_n\}_{n=1}^N$ of $\mathfrak{W}$ established in the previous section:

$$c_k = \boldsymbol{w}^T \boldsymbol{b}_k; \tag{7}$$

$$d_k = \mathcal{P}_{\mathfrak{W}}(\boldsymbol{e}_T)^T \boldsymbol{b}_k. \tag{8}$$

Next, we weaken (realized by subtraction) all components of $\boldsymbol{w}$ and then add back the projected text $\mathcal{P}_{\mathfrak{W}}(\boldsymbol{e}_T)$, with a coefficient such that the sum of coefficients is preserved so that the embedding does not deviate too much:

$$\mathcal{A}_{\mathfrak{W},T}(\boldsymbol{w}, \alpha) := \sum_{k=1}^N \left(c_k - \alpha\,|c_k|\right) \boldsymbol{b}_k + \frac{\alpha \sum_{k=1}^N |c_k|}{\sum_{k=1}^N d_k} \mathcal{P}_{\mathfrak{W}}(\boldsymbol{e}_T), \tag{9}$$

where $\alpha \in \mathbb{R}^+$ controls the augmenting power, contributing to a controllable latent manipulation, demonstrated in Figure 5b. The two terms in Eq. (9) are equivalent to weakening the components that are small in the projected text.

We also include two more options for projection and three more for augmentation in Appendix A.2, including the case when $\mathfrak{W}$ is not a linear subspace but a general manifold (so that there is no notion of basis). Future researchers can also develop more effective options suitable for their own specific tasks, and this is where lies the extensibility of our PAE. A preliminary criteria for evaluation and selection of different PAEs are present in Appendix A.3.

## 5 EXPERIMENT

As a case study, we utilize PAE in a series of text-guided semantic face editing experiments, where we manipulate a high-level (emotion) and a low-level (hairstyle) facial attribute (Patashnik et al., 2021; Lyons et al., 2000). We also show the results of manipulating other facial attributes such as the size of the eyes and the size of mouth in Appendix A.6, and the results of manipulating non-facial images such as dogs in Appendix A.7.

### 5.1 DATA AND PROCEDURE

Given randomly generated latent codes, we generate facial images using an implementation of Style-GAN2 (Karras et al., 2020) with an adaptive discriminator augmentation (ADA) and the pre-trained model weights for the FFHQ face dataset (Karras et al., 2017). We randomly generate initial latent codes, and then filter out invalid faces, leaving 28 images for the task.

As explained in Section 4, PAE can be integrated into any CLIP-based text-guided image manipulation algorithms to improve performance. We integrate PAE into StyleMC (Kocasari et al., 2022), StyleCLIP (Patashnik et al., 2021), and TediGAN (Xia et al., 2021), comparing their performance with and without PAE. To further demonstrate the disentangling power of our method and ablate other influencing factors, we additionally apply PAE to a naive image manipulation algorithm — we directly update the latent code of the randomly generated input images before the differentiable StyleGAN2 generator, according to either the naive loss (Figure 4a) or a loss with PAE (Figure 4b). Note that this approach does not include any additional supervision losses (such as an identity loss (Deng et al., 2019)) or other trainable parameters. As we will see in the following two subsections, PAE can achieve identity preservation even without the identity loss.

In total, we compare four pairs of models; each pair consists of the original version (Figure 4a) *vs.* its +PAE version (Figure 4b): 1. Naive approach (Nv) *vs.* Naive+PAE (Nv+); 2. StyleMC (SM) (Kocasari et al., 2022) *vs.* StyleMC+PAE (SM+); 3. StyleCLIP (SC) (Patashnik et al., 2021) *vs.*

StyleCLIP+PAE (SC+); and 4. TediGAN (TG) TediGAN (Xia et al., 2021) *vs*. TediGAN+PAE (TG+). Note that each pair naturally forms an ablation study for PAE. The improvement brought by the PAE can be observed by comparing each original model with its +PAE version.

To construct the PAEs, for emotion editing, we use $\mathcal{P}_{\text{emotion}}^{\text{GS}}$, defined via six basic human emotions (Ekman, 1992) — *happy*, *sad*, *angry*, *fearful*, *surprised*, and *disgusted* — as the semantic basis; for hairstyle editing, we use $\mathcal{P}_{\text{hairstyle}}^{\text{PCA}}$ defined via 68 hairstyle texts and ten principal components. The augmenting power is automatically selected according to the criteria stated in Appendix A.3.

## 5.2 QUALITATIVE COMPARISON

A qualitative comparison is shown in Figure 5a. The text prompts are written to the left of each row. Furthermore, in Figure 5b, we fix the text prompt to be "happy" and vary the augmenting power $\alpha$, showing the controllability of our PAE. Please refer to Appendix A.6 for full results.

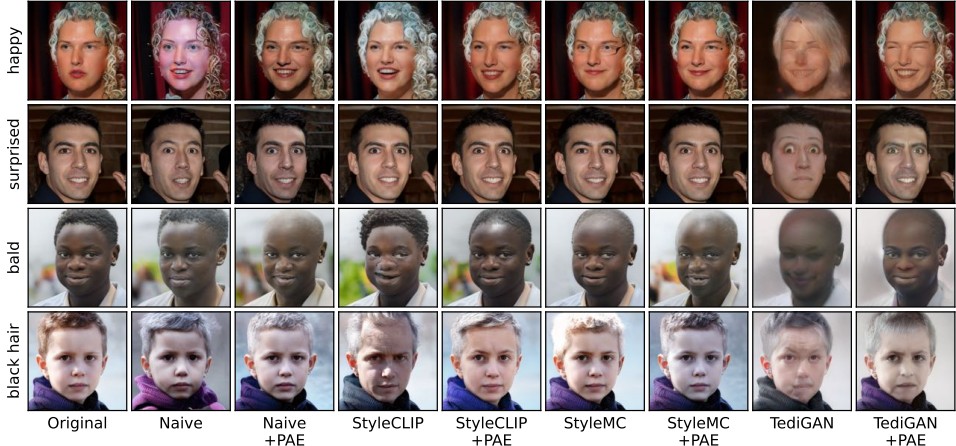

(a) Comparison of eight models in text-guided face editing. We see that PAE promotes disentangled editing.

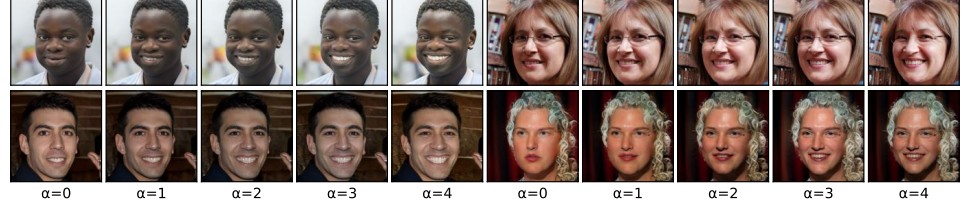

(b) Varying the augmenting power $\alpha$ when making faces happy to show the controllability

Figure 5: Experiment results

From the Figure, we can see that in the naive approach, changes are not made in a disentangled way. For example, the hair color and the lighting condition of the face in the first row changed; the identity of the face in the second row changed; *etc*. These problems are also present in the other three original models. We can also see that some edits do not have high quality or accuracy. Comparing models before and after equipping with PAE, we see that PAE enables a much more disentangled, realistic and accurate face manipulation.

## 5.3 QUANTITATIVE COMPARISON

In Table 1, we measure the performance of the aforementioned eight models with a quantitative comparison in seven metrics: Fréchet Inception Distance (FID) (Heusel et al., 2017) (measures the quality of manipulated images), Learned Perceptual Image Patch Similarity (LPIPS) (measures the perceptual similarity of manipulated images to the original ones), Identity loss (IL) (uses ArcFace (Deng et al., 2019) to measure the degree of facial identity preservation), disentanglement measurement with a facial-attribute classifier (Dis-C) (uses a facial attribute classifier (Serengil & Ozpinar, 2021) to evaluate the degree of disentanglement by measuring changes in the model classification of

irrelevant attributes before and after editing), accuracy measurement with a facial-attribute classifier (Acc-C) (uses the same facial attribute classifier to measure the degree of conformity to the text prompt), disentanglement evaluation from survey (Dis-S) (measures the degree of disentanglement with a survey), and accuracy evaluation from survey (Acc-S) (measures the degree of conformity to the text prompt with a survey). The first three scores are computed over the editing result of 5880 randomly generated images. The middle two scores from classifiers are computed over 48 emotion editing images: Dis-C is calculated from the percentage of the model classification of irrelevant attributes (age and race) that remain unchanged after the manipulation; and Acc-C is calculated from the percentage of output images whose model-predicted emotion is the same as the text prompt. Finally, the two scores from survey are obtained from a user study involving 50 participants of various backgrounds evaluating the editing of 36 randomly generated images. We used 7.0 for $\alpha$ in PAE in all methods.

Table 1: Seven metrics for eight image editing models. The ↓ besides a metric means that a lower score is better and the ↑ means the opposite.

| Model | Nv | Nv+ | SC | SC+ | SM | SM+ | TG | TG+ |
|---|---|---|---|---|---|---|---|---|
| FID ↓ | 96.322 | **73.061** | 75.684 | **67.800** | 73.522 | **63.161** | 79.784 | **63.203** |
| LPIPS ↓ | 0.251 | **0.103** | 0.157 | **0.071** | 0.123 | **0.094** | 0.298 | **0.073** |
| IL ↓ | 0.279 | **0.140** | 0.198 | **0.134** | 0.144 | **0.119** | 0.618 | **0.232** |
| Dis-C (%) ↑ | 30.2 | **36.3** | **39.6** | 35.4 | 33.3 | **38.6** | 30.2 | **35.5** |
| Acc-C (%) ↑ | 20.8 | **60.4** | 22.9 | **58.3** | 29.2 | 27.1 | 58.3 | **62.5** |
| Dis-S (%) ↑ | 5.2 | **18.3** | 5.6 | **18.8** | 6.5 | **20.2** | 1.5 | **13.1** |
| Acc-S (%) ↑ | 3.8 | **19.7** | 8.7 | **24.7** | 7.8 | **23.8** | 6.8 | **20.8** |

From Table 1, we see that Nv+ already outperforms the four original models in all scores except for Dis-C, where it is second to SC. More importantly, we see that almost all scores of the four models improved after equipping with PAE. It is also worth noting that IL is included in the training loss in SC and SM but not in Nv+, nevertheless Nv+ outperforms SC and SM in terms of IL. These observations strongly indicate that PAE can improve the image quality, identity preservation, disentanglement and accuracy in image editing tasks.

## 6    DISCUSSION, LIMITATION, AND FUTURE WORK

An interesting observation from our experiment in hairstyle editing is that the difficulty of manipulation complies with the real-world situations. For example, all models find it very hard to change women's hairstyle to be bald whereas men are very easy to lose hair. This suggests that the performance of such image manipulation methods is limited by the real-world datasets that our upstream models (StyleGAN2 and CLIP) are trained on.

Another limitation of our method is that the concrete projection and augmentation operations need manual selection for the best result. We automated a coarse selection by the criteria introduced in Appendix A.3, but a rigorously defined, numeric metric for disentanglement, accuracy, *etc*. could benefit the performance. Also, texts that does not associate with an obvious attribute (*e.g.*, a celebrity's name) are harder to project to the subspace.

## 7    CONCLUSION

In this paper, we considered the problem of CLIP-based text-guided image manipulation. We introduced a novel projection-augmentation embedding (PAE) to facilitate a more disentangled, interpretable and controllable image editing. PAE is a general technique that can be seamlessly integrated into any other CLIP-based models; it is simple to use and has the advantage of compute efficiency, extensibility and generalizability. We demonstrated its effectiveness in a text-guided semantic face editing experiment at both high-level (emotion) and low-level (hairstyle) editing. We integrate PAE with other state-of-the-art CLIP-based face editing models, showing the improvements brought by PAE in various perspectives.

## 8 REPRODUCIBILITY STATEMENT

The results in the main part and the appendix of this paper can be reproduced using the source code submitted as supplementary material together with this paper. The project web page as well as the repository will be made publicly available after the blind review process.

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

# A APPENDIX

## A.1 SIMILARITY OF DIFFERENT CLIP EMBEDDINGS

As a complement to Figure 2, we record the averaged cosine similarity of the embeddings from different modalities in Table 2. We see that image embeddings and text embeddings lie in different regions in $\mathfrak{C}$ and have lower inter-modality similarity compared to the intra-modality similarity, regardless of their semantic meanings. From the Table we can also see that PAEs have higher cosine similarity to images (0.491 and 0.493) than texts to images (0.200).

Table 2: Averaged cosine similarity of different CLIP embeddings. Note that image embeddings and text embeddings have lower inter-modality similarity. Also note that PAEs have higher intra-modality similarity (lower variation) than images, showing that they discard some noisy information.

|  | face text | emotion text | hair text | face image | $\mathcal{E}^{\mathrm{PCA}}_{\mathrm{emotion}}$ | $\mathcal{E}^{\mathrm{PCA}}_{\mathrm{hairstyle}}$ |
|---|---|---|---|---|---|---|
| face text | 0.852 | 0.862 | 0.779 | 0.200 | 0.684 | 0.630 |
| emotion text |  | 0.877 | 0.784 | 0.198 | 0.693 | 0.633 |
| hair text |  |  | 0.764 | 0.201 | 0.634 | 0.622 |
| face image |  |  |  | 0.567 | 0.491 | 0.493 |
| $\mathcal{E}^{\mathrm{PCA}}_{\mathrm{emotion}}$ |  |  |  |  | 0.771 | 0.731 |
| $\mathcal{E}^{\mathrm{PCA}}_{\mathrm{hairstyle}}$ |  |  |  |  |  | 0.725 |

Note that this discrepancy of image embeddings and text embeddings in the CLIP space is general and not restricted to faces. We plot the embeddings of 100 random images and 100 random texts visualized using PCA in Figure 6 and also record the similarity in Table 3. We note the same observation that image embeddings and text embeddings lie in different regions in $\mathfrak{C}$ and have lower inter-modality similarity compared to the intra-modality similarity.

Table 3: Averaged cosine similarity of CLIP embeddings of 100 random images and texts. Note that image embeddings and text embeddings have lower inter-modality similarity.

|  | image | text |
|---|---|---|
| image | 0.664 | 0.199 |
| text |  | 0.845 |

## A.2 MORE OPTIONS FOR PROJECTION AND AUGMENTATION

In this section we present two more options for projection and three more for augmentation. About notations: we give each option a short identifier (or not at all) and add it to the superscript of $\mathcal{P}_{\mathfrak{W}}$, $\mathcal{A}_{\mathfrak{W},T}$ or $\mathcal{E}_{\mathfrak{W}}$. For example, $\mathcal{E}^{\mathrm{GS,+}}_{\mathfrak{W}}$ means the projection-augmentation embedding with a projection onto an orthonormal basis ($\mathcal{P}^{\mathrm{GS}}_{\mathfrak{W}}$) and an augmentation that preserves the coefficients ($\mathcal{A}^{+}_{\mathfrak{W},T}$).

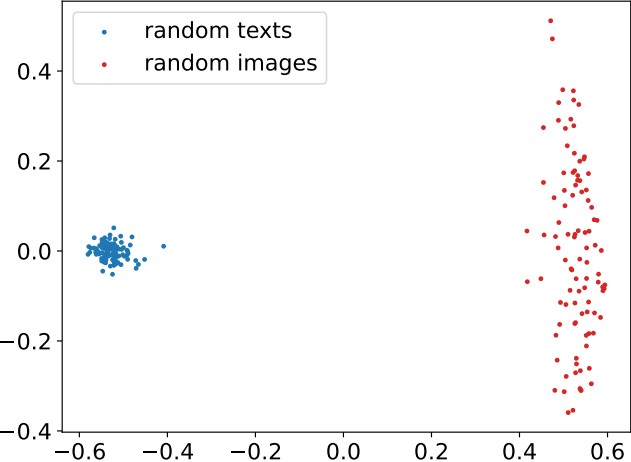

Figure 6: PCA visualization of 100 random images and 100 random texts in the CLIP space. Note that image and text embeddings lie in differet regions.

### A.2.1 PROJECTION

A simpler type of projection, $\mathcal{P}_{\mathfrak{W}}$ also assumes $\mathfrak{W}$ to be a linear subspace of the CLIP space $\mathfrak{C}$ and is very similar to $\mathcal{P}_{\mathfrak{W}}^{\text{GS}}$ except that it does not orthonormalize the basis vectors.

$$\mathcal{P}_{\mathfrak{W}}(\boldsymbol{e}_I) := \sum_k \left(\boldsymbol{b}_k^T \boldsymbol{e}_I\right) \boldsymbol{b}_k. \tag{10}$$

Note that in this case, the dot product with the basis vectors is not a strict projection onto a linear subspace. However, this simple option also worked well in our prior experiments. This is possibly due to the high dimensionality of the CLIP space (512): if we do not have many basis vectors (*e.g.*, $< 20$), their pairwise dot products tend to be very small and thus are nearly orthogonal to each other already.

The second option, $\mathcal{P}_{\mathfrak{W}}^{\text{All}}$, does not assume that $\mathfrak{W}$ is necessarily a linear subspace. Instead, it can be any manifold. $\mathcal{P}_{\mathfrak{W}}^{\text{All}}$ directly stores the set $\mathcal{T}$. By storing embeddings of **all** related texts, it is effectively sampling and storing the points on the manifold $\mathfrak{W}$, and when performing the projection onto a surface, the best approximation would be to pick the point on the surface that is closest to the vector to be projected:

$$\mathcal{P}_{\mathfrak{W}}^{\text{All}}(\boldsymbol{e}_I) := \underset{\boldsymbol{e} \in \mathcal{T}}{\arg\min} \operatorname{CosSim}(\boldsymbol{e}_I, \boldsymbol{e}). \tag{11}$$

Naturally, if we obtain a larger $\mathcal{T}$, we sample more points from the manifold and better approximate the projection.

### A.2.2 AUGMENTATION

We propose three more option for the augmentation operation: $\mathcal{A}_{\mathfrak{W},T}$, $\mathcal{A}_{\mathfrak{W},T}^{\text{Ex}}$ and $\mathcal{A}_{\mathfrak{W},T}^{\text{ExD}}$. The first one is used with projections $\mathcal{P}_{\mathfrak{W}}^{\text{GS}}$, $\mathcal{P}_{\mathfrak{W}}$ and $\mathcal{P}_{\mathfrak{W}}^{\text{PCA}}$, and the last two with $P^{\text{All}}$.

The simplest $\mathcal{A}_{\mathfrak{W},T}$ adds the text and image together, with a coefficient $\alpha$ controlling the augmenting power:

$$\mathcal{A}_{\mathfrak{W},T}(\boldsymbol{w}) := \boldsymbol{w} + \alpha \boldsymbol{e}_T. \tag{12}$$

However, simply adding up $\boldsymbol{e}_T$ may result in a vector too similar to the original embedding, likely resulting in an optimization process barely doing anything. In that case, the augmentation introduced in equation 9 is a better option because it additionally weakens a certain amount on other components (while preserving the sum of coefficients). To avoid the misunderstanding, we denote by $\mathcal{A}_{\mathfrak{W},T}$ this simplest augmentation and denote by $\mathcal{A}_{\mathfrak{W},T}^{+}$ the augmentation in equation 9.

The last two types of augmentation, $\mathcal{A}_{\mathfrak{W},T}^{\text{Ex}}$ and $\mathcal{A}_{\mathfrak{W},T}^{\text{ExD}}$, are for $P^{\text{All}}$. Since $P^{\text{All}}$ gives an embedding $\boldsymbol{e} \in \mathcal{T}$, which is expected to indicate the current feature of the image, if we want to change that

feature to be the one specified by the text $T$, we can simply **ex**change $e$ with the text embedding $e_T$:

$$\mathcal{A}^{\text{Ex}}_{\mathfrak{W},T}(w) := w + \alpha \left( e_T - \mathcal{P}^{\text{All}}_{\mathfrak{W}}(e_I) \right), \quad (13)$$

where $\alpha$ is the augmenting power.

The final $\mathcal{A}^{\text{ExD}}_{\mathfrak{W},T}$ is a more robust version than $\mathcal{A}^{\text{Ex}}_{\mathfrak{W},T}$. Instead of doing the one-for-one exchange as above, we can do one-for-$\alpha$ exchange. More precisely, we still add in $\alpha$ copies of $e_T$, but instead of subtracting $\alpha$ copies of the most similar text embeddings, we subtract the $\alpha$ **d**ifferent, most similar text embeddings (each embedding only subtracted once).

Naturally, between the one-for-one exchange in $\mathcal{A}^{\text{Ex}}_{\mathfrak{W},T}$ and one-for-$\alpha$ exchange in $\mathcal{A}^{\text{ExD}}_{\mathfrak{W},T}$, there exist many other options from one-for-$k$ for any $k$ between 1 and $\alpha$. We leave this exploration to future work.

## A.3 EVALUATION CRITERIA

In the future other researchers may want to develop new types of PAE for their specific tasks. In this section we provide two straightforward criteria to evaluate and select different options of PAE:

1. We need $\text{CosSim}(\mathcal{E}_{\mathfrak{W}}, e_{I_t})$ to be as large as possible, where $e_{I_t}$ is the embedding of the target image, *i.e.* the ideal target. This is because our goal is to approximate the inaccessible $e_{I_t}$. At least, we need $\text{CosSim}(\mathcal{E}_{\mathfrak{W}}, e_{I_t}) > \text{CosSim}(e_T, e_{I_t})$ so that we gain something by replacing $e_T$ with $\mathcal{E}_{\mathfrak{W}}$ as an optimization target.
2. We need $\text{CosSim}(\mathcal{E}_{\mathfrak{W}}, e_{I_t}) > \text{CosSim}(\mathcal{E}_{\mathfrak{W}}, e_I)$, otherwise PAE is more similar to the original image and the no change towards $e_{I_t}$ will be made during the optimization.

We evaluate all the eight options of PAE (two proposed in Section 4.1 and six in Appendix A.2) in the text-guided facial emotion editing experiment described in Section 5. $\mathcal{P}^{\text{GS}}_{\mathfrak{W}}$ is realized by using six basic human emotions (Ekman, 1992) as a semantic basis:

"happy", "sad", "angry", "fearful", "surprised", "disgusted";

$\mathcal{P}^{\text{PCA}}_{\mathfrak{W}}$ and $\mathcal{P}^{\text{All}}_{\mathfrak{W}}$ are implemented with a corpus $\mathcal{T}$ consisting of 277 emotion texts found online and ten principal components. We plot the above quantities *vs.* $\alpha$ in Figure 7. Each color corresponds to an option of $\mathcal{E}_{\mathfrak{W}}$. Restating the above criteria, within each color, we need 1. the solid line to be as large as possible and at least be above the orange horizontal line; and 2. the dash-dotted line to be above the black horizontal line. We see that in this particular experiment of facial emotion editing, it is more suitable to use $\mathcal{E}^{+}_{\mathfrak{W}}$ with an $\alpha \in [5, 15]$, $\mathcal{E}^{\text{GS},+}_{\mathfrak{W}}$ with an $\alpha \in [10, 15]$, or $\mathcal{E}^{\text{PCA},+}_{\mathfrak{W}}$ with an $\alpha \in [2.5, 5]$. Note that as the $+$ versions perturb the embedding more, the dash-dotted lines for $\mathcal{E}^{+}_{\mathfrak{W}}$, $\mathcal{E}^{\text{GS},+}_{\mathfrak{W}}$ and $\mathcal{E}^{\text{PCA},+}_{\mathfrak{W}}$ are higher than their non-$+$ counterparts.

## A.4 A FAILED CASE: DOUBLE-PROJECTED EMBEDDING

This section presents the double projected embedding (DPE) that has similar idea of using CLIP subspaces to extract relevant information from the embeddings. DPE is simpler than PAE in that it does not have the second step of augmenting the projected image embedding; instead, it projects both image and text embeddings onto the subspace and directly optimizes the image towards the text in the subspace.

We conduct the same text-guided emotion editing experiment as in Section 5 using DPE$^{\text{GS}}$ and DPE$^{\text{PCA}}$ with 20 principal components. The results are summarized in Figure 8 and Figure 9, respectively.

We can see from the results that although most faces indeed changed the emotion according to the text prompts, we lost the disentanglement: in some manipulation, the lighting condition, the background, the hair, or even the facial identity changed.

One possible explanation is that by projecting faces onto the emotion subspace, most information except the emotion has been lost, so we cannot preserve irrelevant attributes. Recall that in constructing PAE, the final step is to add back the residual $r$ to go back to the "image region" (see Eq. (4)). We hypothesize that this $r$ exactly contains the irrelevant information that we need to preserve. This example also shows that the necessity and importance of using an optimization target that is close enough to the embedding $e_I$ of the original image.

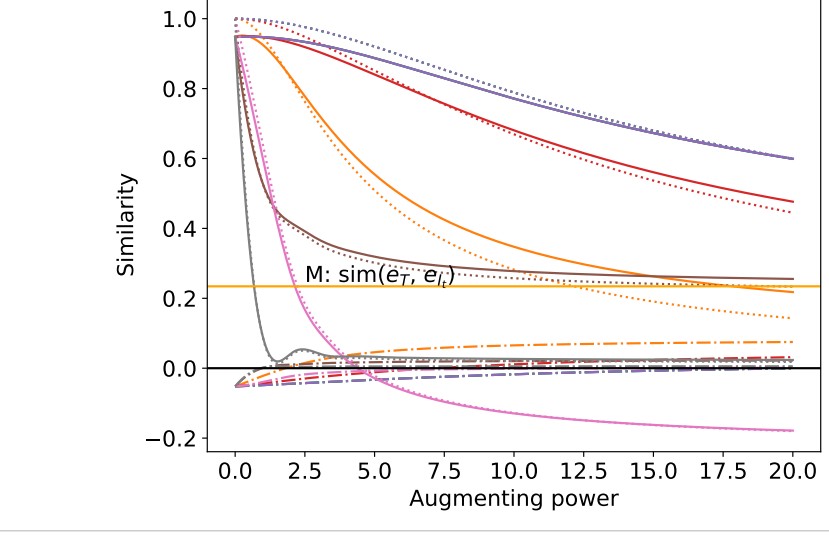

Figure 7: Comparison of eight options of PAE in a text-guided facial emotion editing experiment. Each option is color-coded. In order for $\mathcal{E}_{\mathfrak{W}}$ to be effective, we need the solid line to be as large as possible and at least be above the orange horizontal line and the dash-dotted line to be above the black horizontal line. In this experiment, the best three options are $\mathcal{E}_{\mathfrak{W}}^{+}$ ($\alpha \in [5, 15]$), $\mathcal{E}_{\mathfrak{W}}^{\text{GS},+}$ ($\alpha \in [10, 15]$) and $\mathcal{E}_{\mathfrak{W}}^{\text{PCA},+}$ ($\alpha \in [2.5, 5]$). Note also that as the $+$ versions perturb the embedding more, the dash-dotted lines for $+$ versions are generally higher than their non-$+$ counterparts.

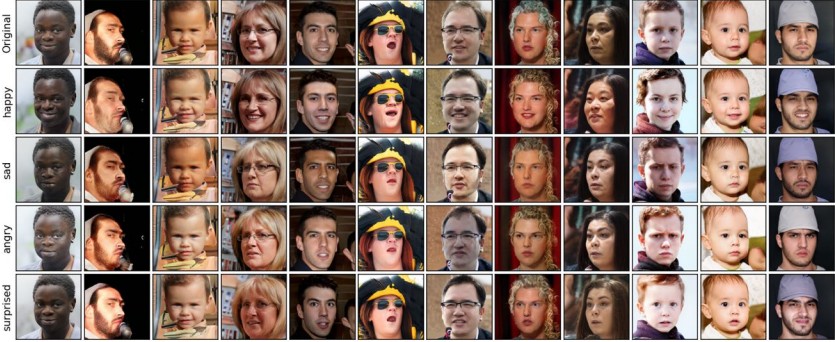

Figure 8: Text-guided emotion editing using $\text{DPE}^{\text{GS}}$. Note that changes are not disentangled.

### A.5 CLIP Subspace Extracts Relevant Information from Texts

In this section we conduct two experiments to demonstrate that our emotion subspace $\mathfrak{W}$ can extract the relevant information from texts. The emotion subspace $\mathfrak{W}$ is created either using $\mathcal{P}_{\mathfrak{W}}^{\text{GS}}$ or $\mathcal{P}_{\mathfrak{W}}^{\text{PCA}}$, but they result in similar observations.

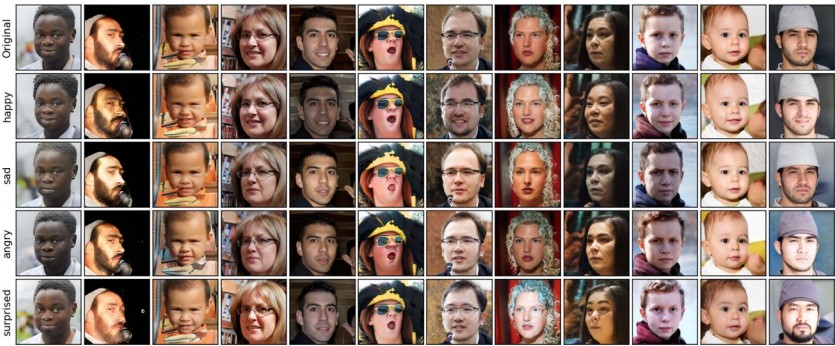

Figure 9: Text-guided emotion editing using DPE$^{\text{PCA}}$. Note that changes are not disentangled.

### A.5.1 EXTRACT EMOTION FROM EMOTION-ANIMAL PHRASES

We compare the averaged similarity of emotion-animal phrases in the original CLIP space $\mathfrak{C}$ and in the emotion space $\mathfrak{W}$ created by $\mathcal{P}_{\mathfrak{W}}^{\text{GS}}$. Each emotion-animal phrase consists of an adjective for emotion (*e.g.*, "happy", "sad") qualifying a noun for animal (*e.g.*, "horse", "man"). If $\mathfrak{W}$ is able to capture the emotion information, then we expect that the similarity of phrases of same animal but different emotions (*e.g.*, "happy horse", "sad horse") is high in $\mathfrak{C}$ (because they are the same animal) but is low in $\mathfrak{W}$ (because the emotions are different). On the contrary, if the emotion is the same but the animals are different (*e.g.*, "happy horse", "happy cat"), then we expect that the similarity is high in $\mathfrak{W}$ but low in $\mathfrak{C}$.

The result is tabulated in Table 4 and coincide with our hypothesis: In $\mathfrak{W}$, emotions dominate the similarity rather than the animals, showing that $\mathfrak{W}$ can extract emotion information from the phrases.

Table 4: Averaged similarity of emotion-animal phrases. In $\mathfrak{W}$, emotions dominate the similarity rather than the animals, showing that $\mathfrak{W}$ can capture emotion information from the phrases.

|  | $\mathfrak{C}$ | $\mathfrak{W}$ |
| --- | --- | --- |
| same animal, different emotions | 0.863 | 0.439 |
| same emotion, different animals | 0.834 | 0.926 |

### A.5.2 EXTRACT EMOTION FROM EMOTION TEXTS

We collect five emotions from each of the 5 groups happy, sad, angry, disgusted, fearful (25 emotions in total) and use heat maps to visualize their pairwise similarity in $\mathfrak{C}$, in the emotion subspaces created by $\mathcal{P}_{\mathfrak{W}}^{\text{GS}}$ and $\mathcal{P}_{\mathfrak{W}}^{\text{PCA}}$. The result is shown in Figure 10. We see that emotions in the same group have higher similarity in both emotion subspaces than in $\mathfrak{C}$.

### A.6 FULL RESULT OF TEXT-GUIDED FACE EDITING

In this section we present the full result of the text-guided face editing that could not be placed in Section 5 due to the space limit. Figures 11 to 13 show the result of emotion, hairstyle and physical characteristic editing, respectively, using different PAEs; The text prompts as well as the type of PAE used are written to the left of each row. Figure 14 to Figure 26 compare the eight models in face editing for different text prompts in the aforementioned three editing categories. Figure 27 shows the controllability of our method by varying the augmenting power.

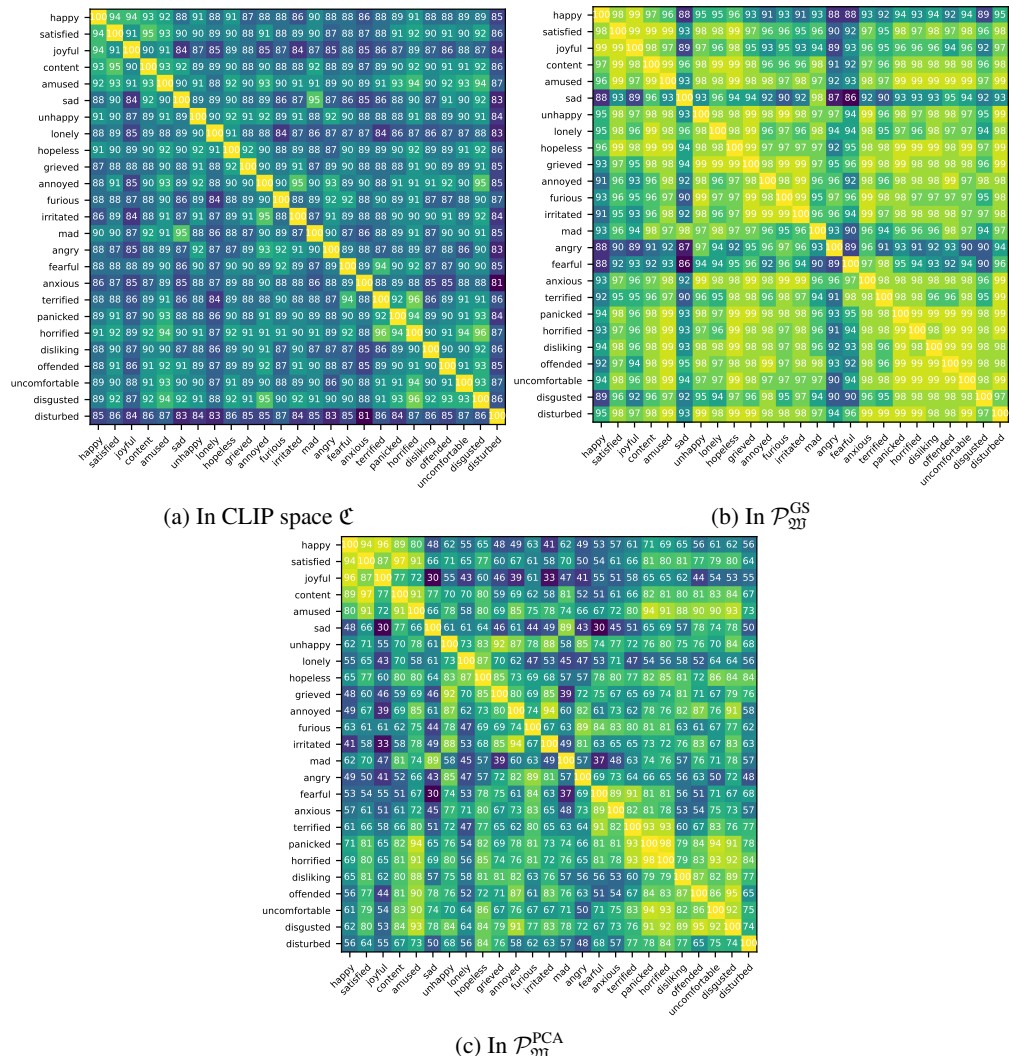

(a) In CLIP space $\mathfrak{C}$

(b) In $\mathcal{P}_{\mathfrak{W}}^{GS}$

(c) In $\mathcal{P}_{\mathfrak{W}}^{PCA}$

Figure 10: Pairwise similarity of 25 emotions in five groups in three different spaces. We see that emotions in the same group have higher similarity in both emotion subspaces than in $\mathfrak{C}$.

## A.7 TEXT-GUIDED EDITING ON NON-FACIAL IMAGES

We also conduct text-guided editing on non-facial images, namely on images of dogs from AFHQ-Dog dataset (Choi et al., 2020). The experiment procedure is very similar to Section 5 except for the three differences:

- StyleGAN2 is pre-trained on the AFHQ-Dog dataset instead of the FFHQ dataset;
- Since StyleCLIP (Patashnik et al., 2021) and TediGAN (Xia et al., 2021) do not release a StyleGAN2 model pre-trained on AFHQ-Dog, we only include the comparison of Naive, Naive+PAE, StyleMC, and StyleMC+PAE;
- The fur subspace is constructed by $\mathcal{P}_{fur}^{PCA}$ defined via extracting principal components from a corpus of nine fur color texts: *white fur, black fur, orange fur, brown fur, gray fur, golden fur, yellow fur, red fur*, and *blue fur*.

Figure 28 shows the aggregated result of Naive+PAE approach. The text prompts are written to the left of each row. Figure 29 to Figure 31 compare the four models for different text prompts.

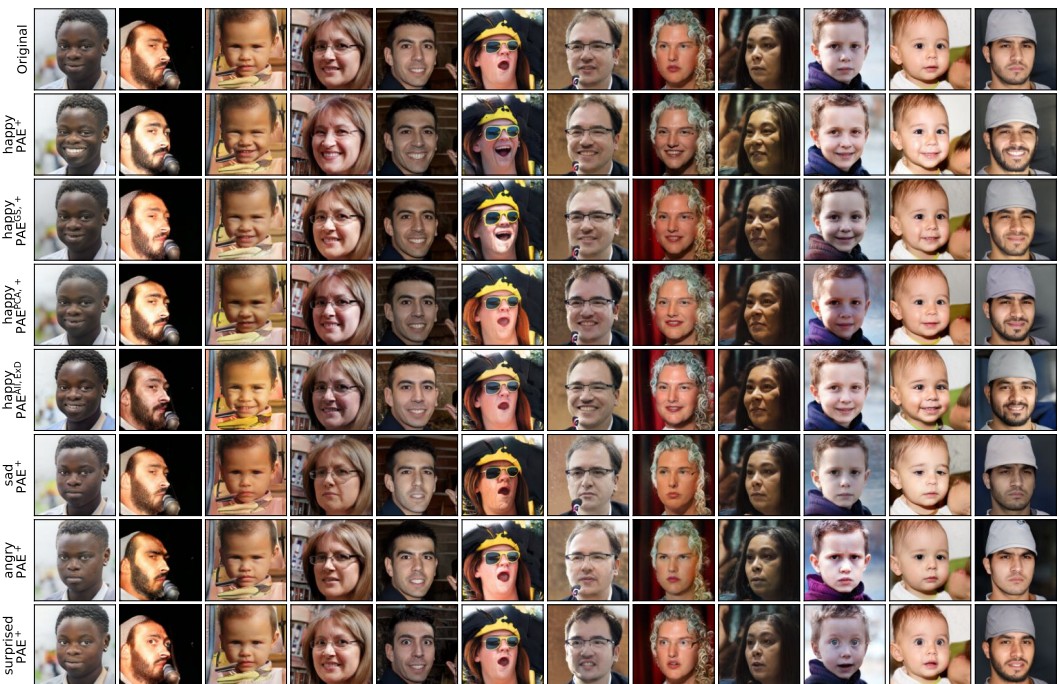

Figure 11: Text-guided emotion editing using different PAEs

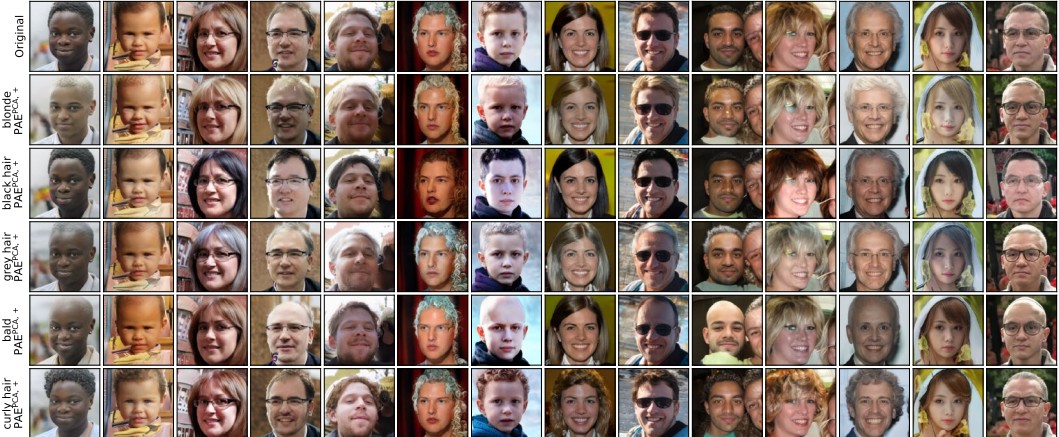

Figure 12: Text-guided hairstyle editing

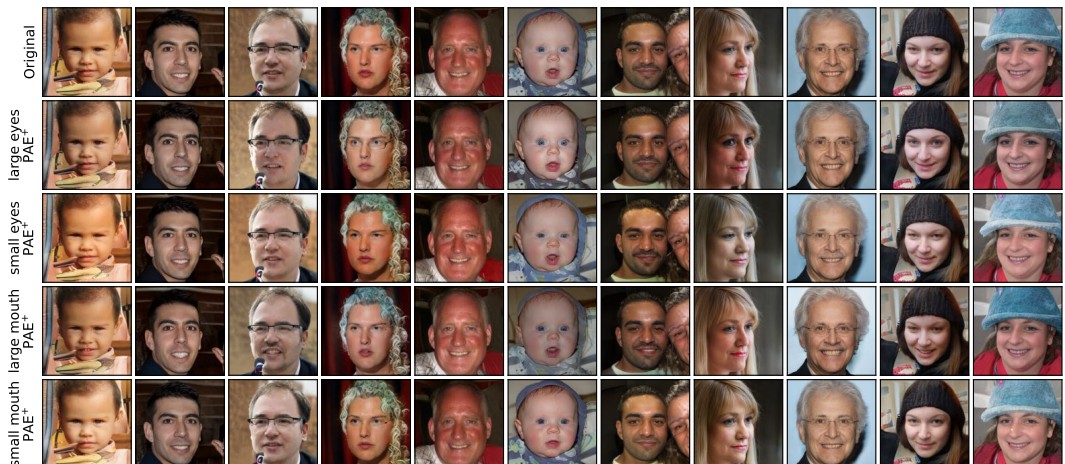

Figure 13: Text-guided physical characteristic editing

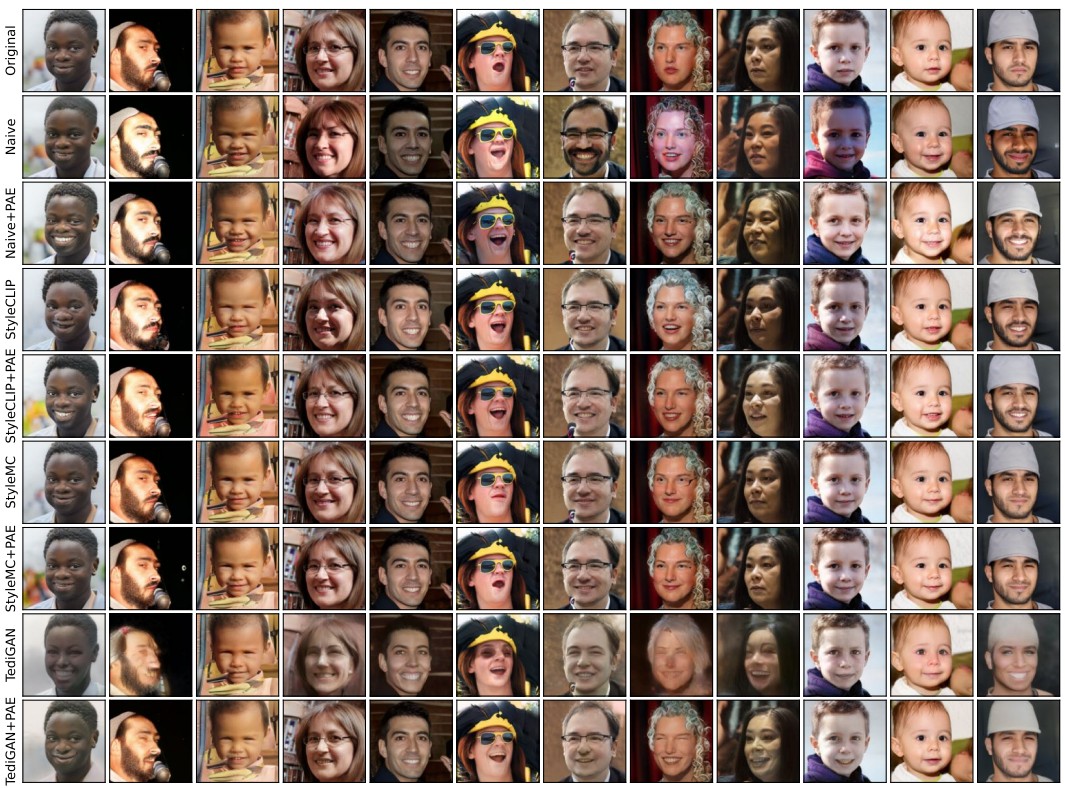

Figure 14: Comparison of different models. The text prompt is "happy".

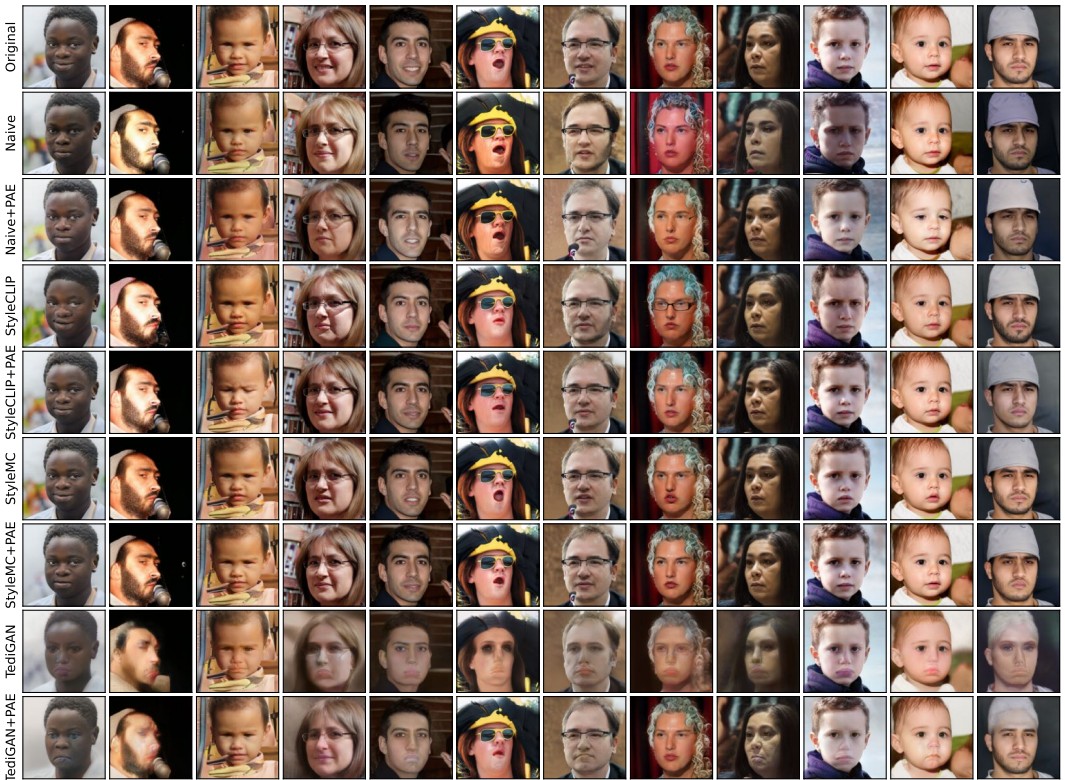

Figure 15: Comparison of different models. The text prompt is "sad".

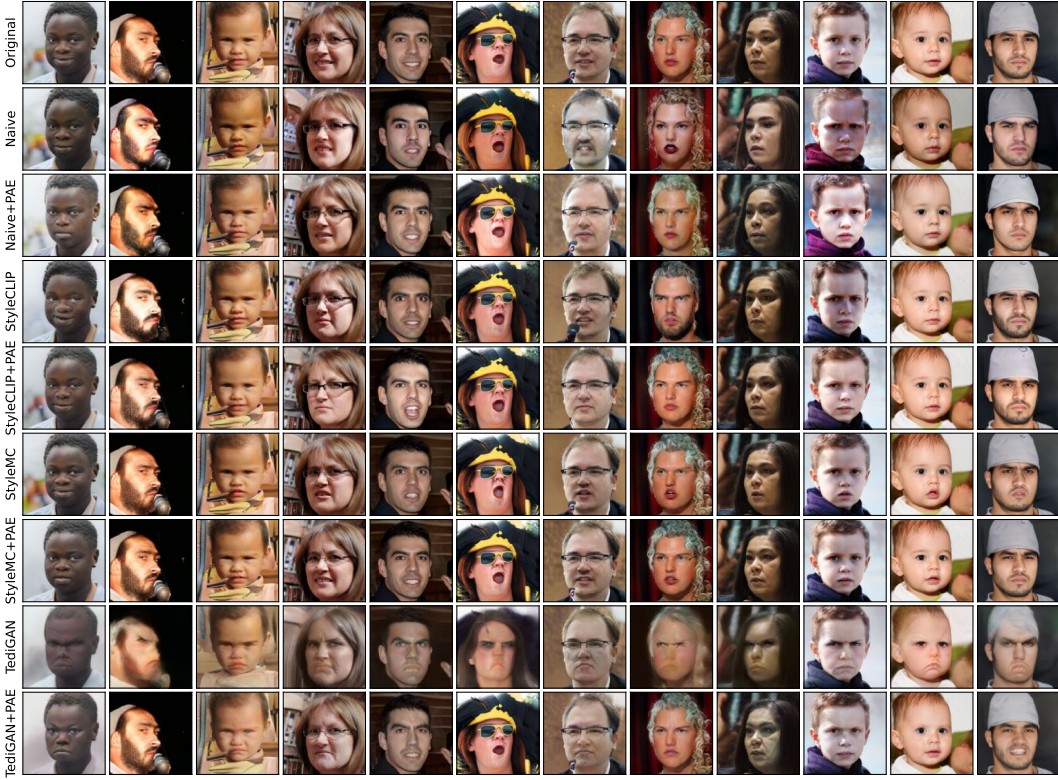

Figure 16: Comparison of different models. The text prompt is "angry".

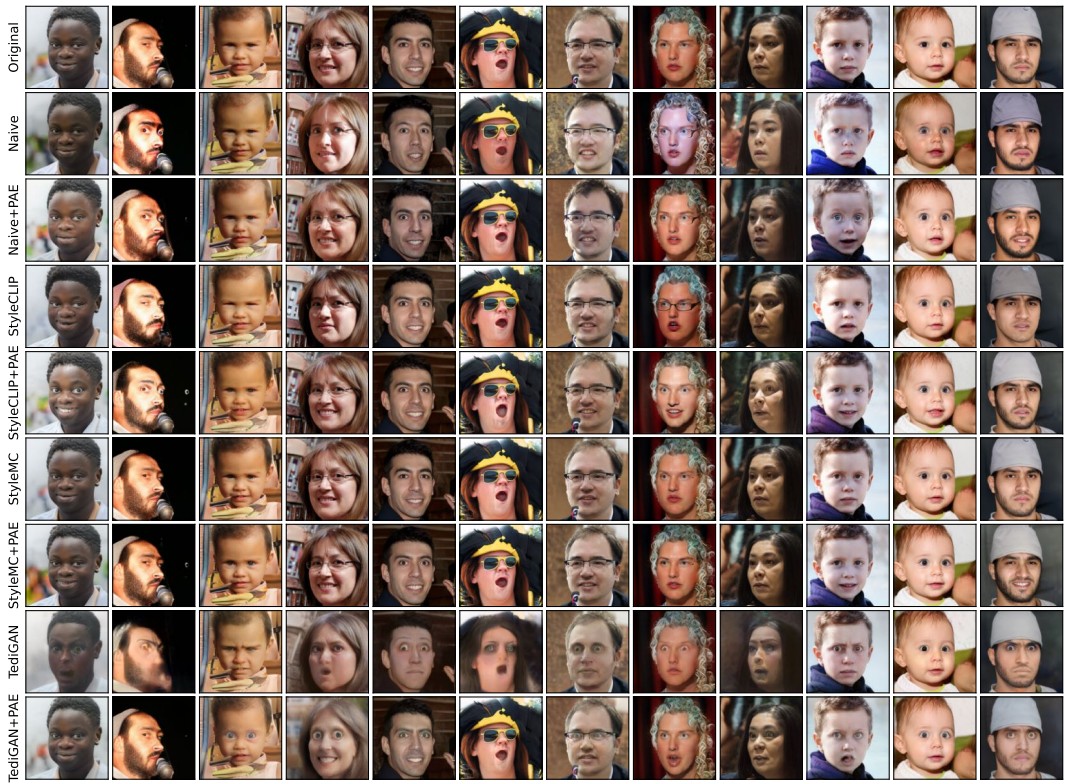

Figure 17: Comparison of different models. The text prompt is "surprised".

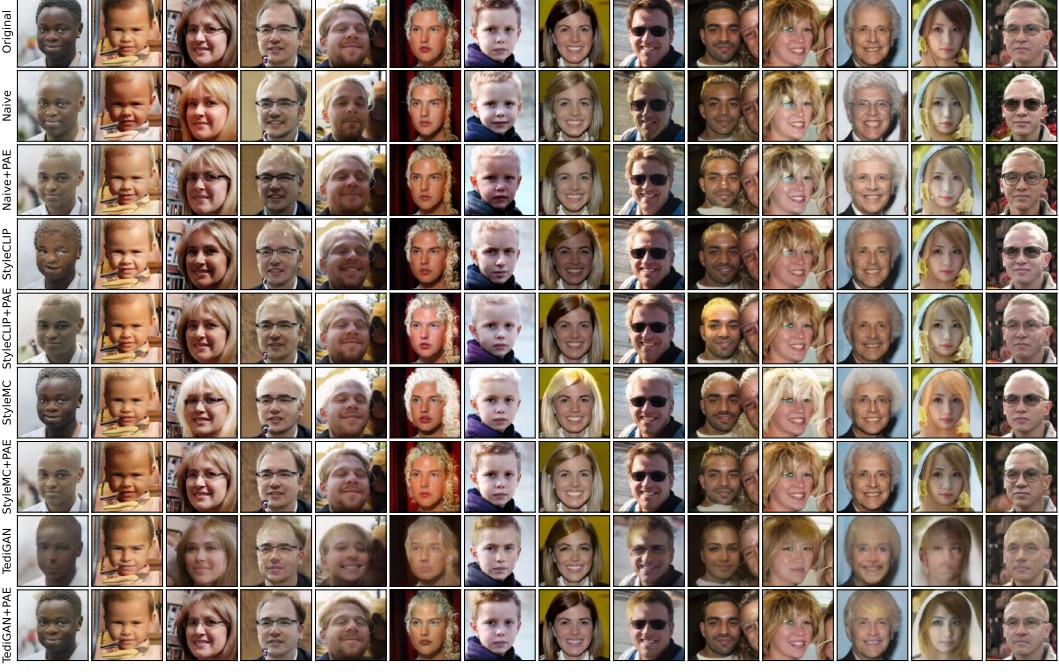

Figure 18: Comparison of different models. The text prompt is "blonde".

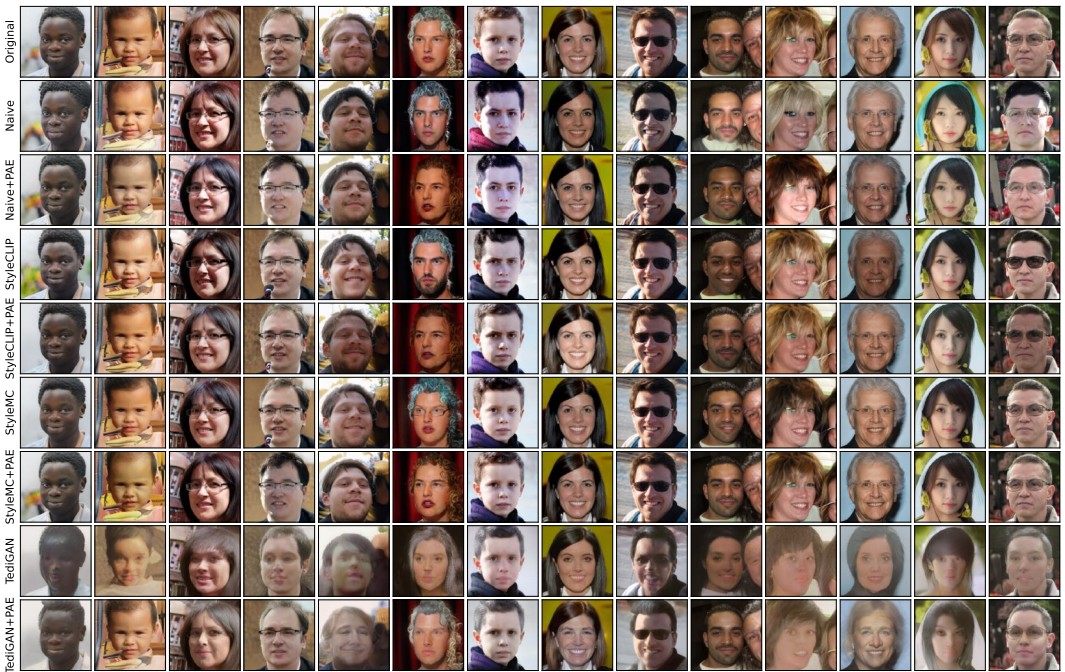

Figure 19: Comparison of different models. The text prompt is "black hair".

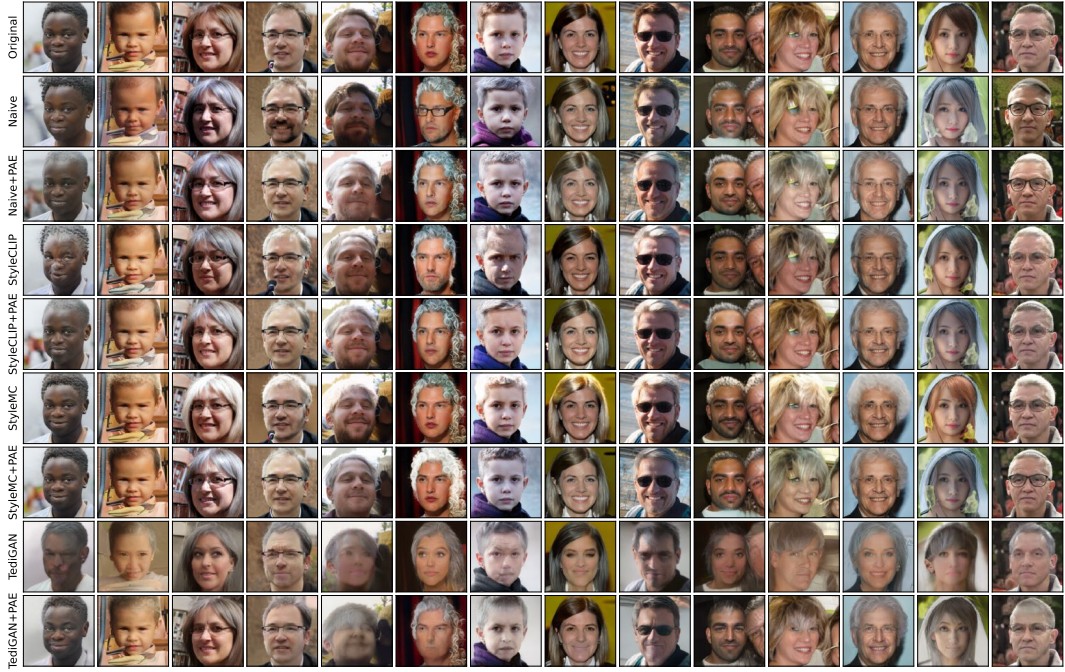

Figure 20: Comparison of different models. The text prompt is "grey hair".

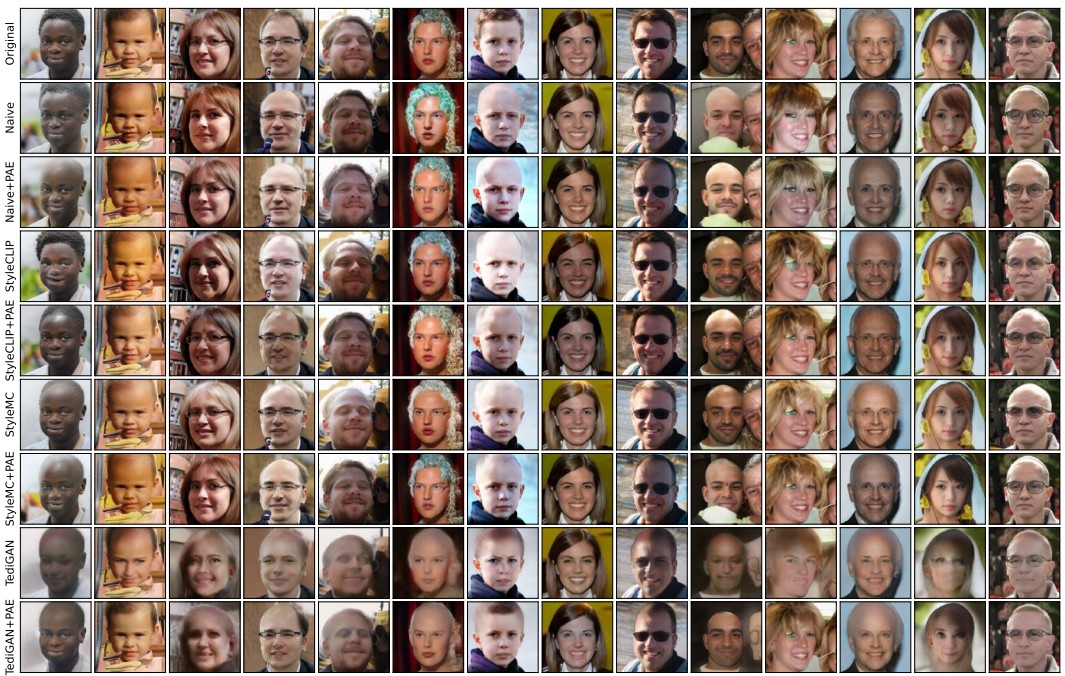

Figure 21: Comparison of different models. The text prompt is "bald".

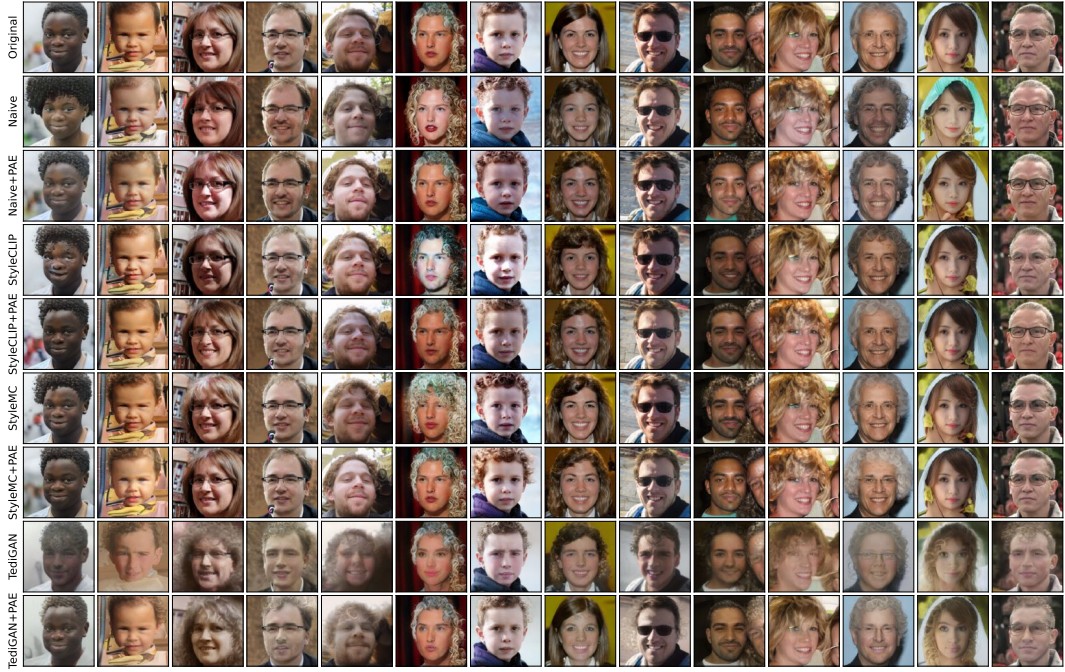

Figure 22: Comparison of different models. The text prompt is "curly hair".

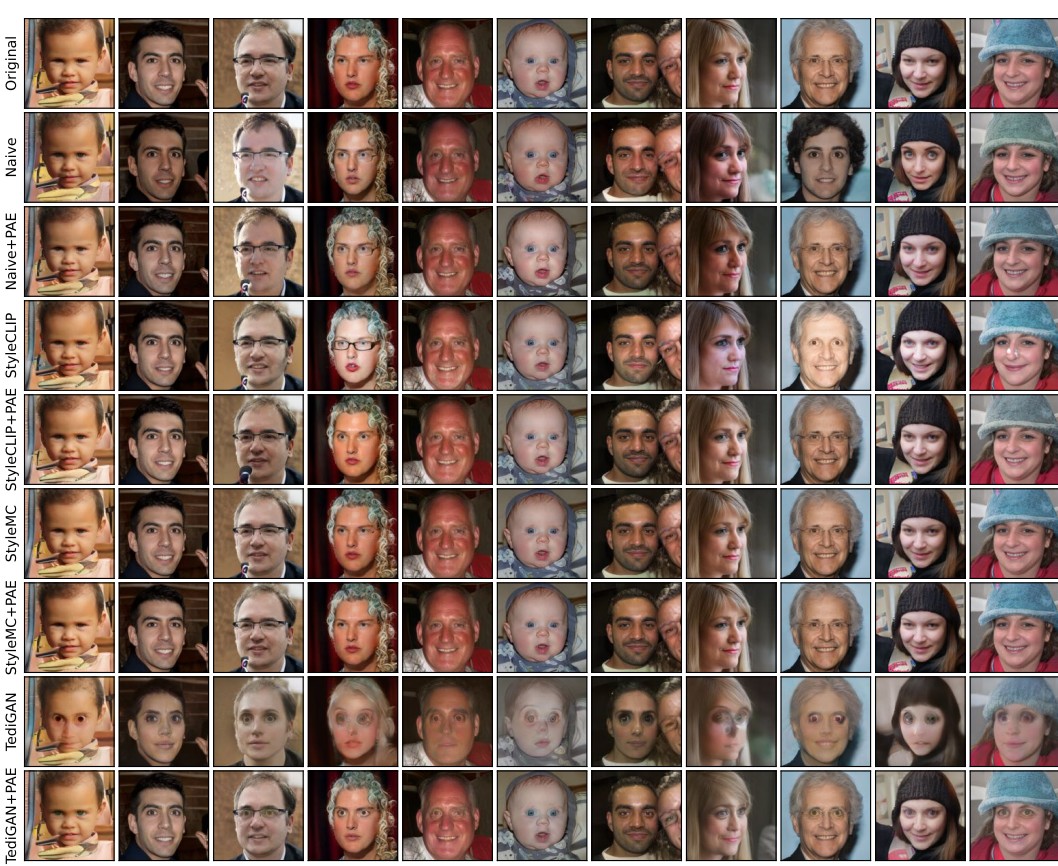

Figure 23: Comparison of different models. The text prompt is "large eyes".

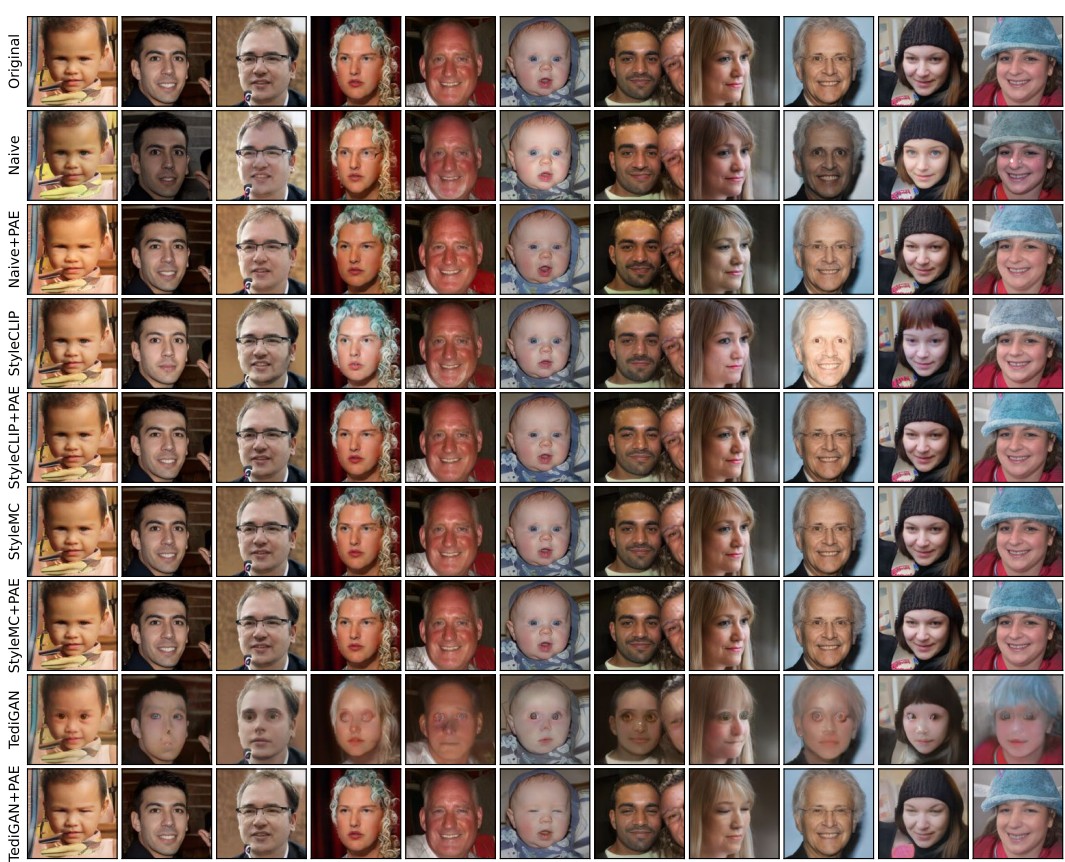

Figure 24: Comparison of different models. The text prompt is "small eyes".

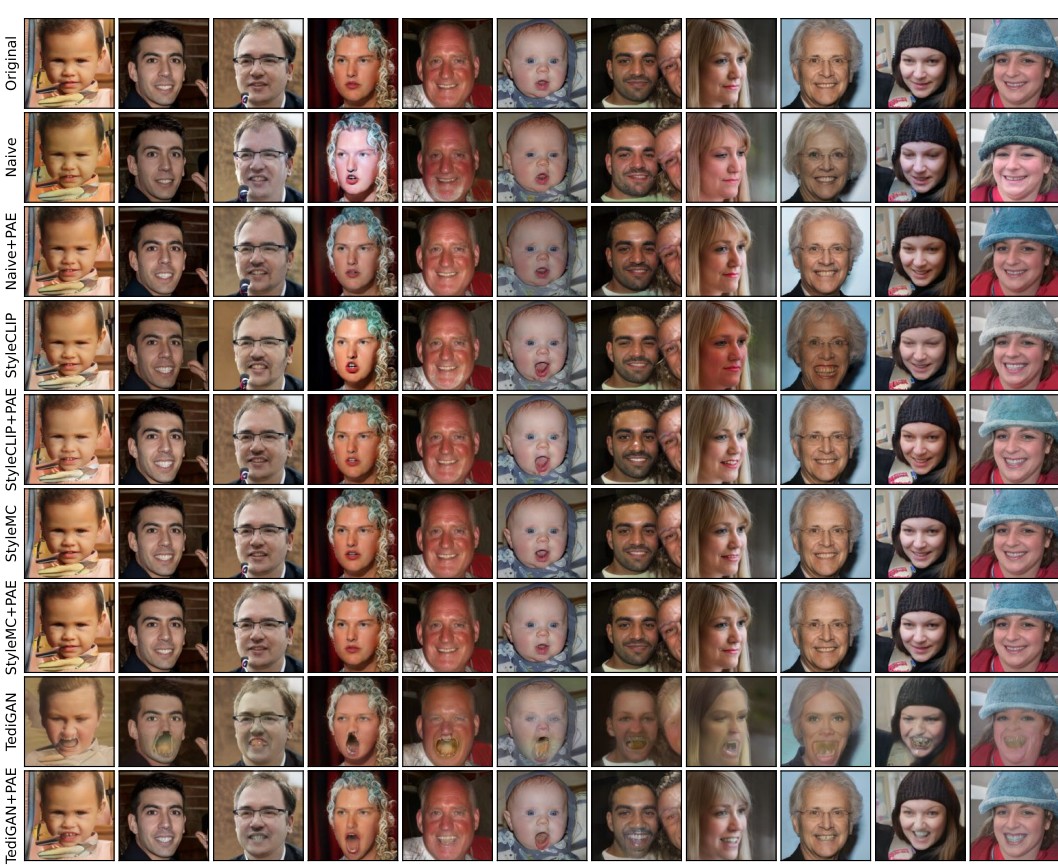

Figure 25: Comparison of different models. The text prompt is "large mouth".

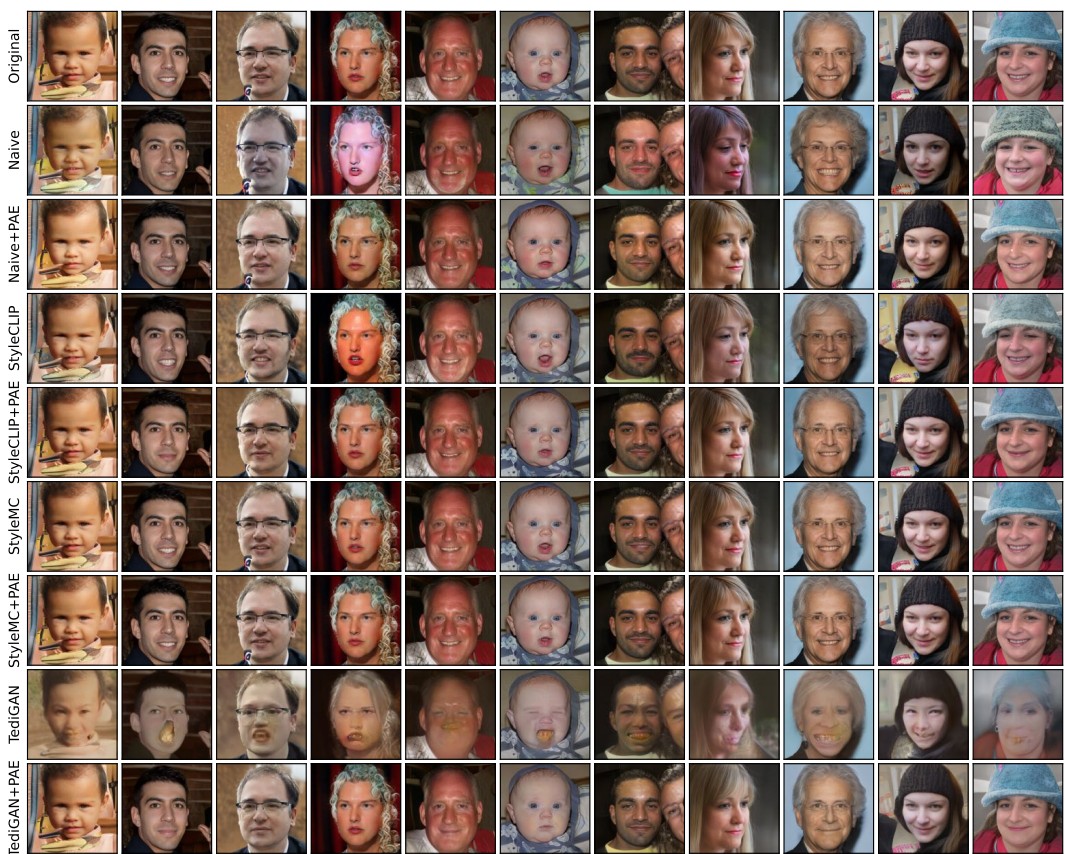

Figure 26: Comparison of different models. The text prompt is "small mouth".

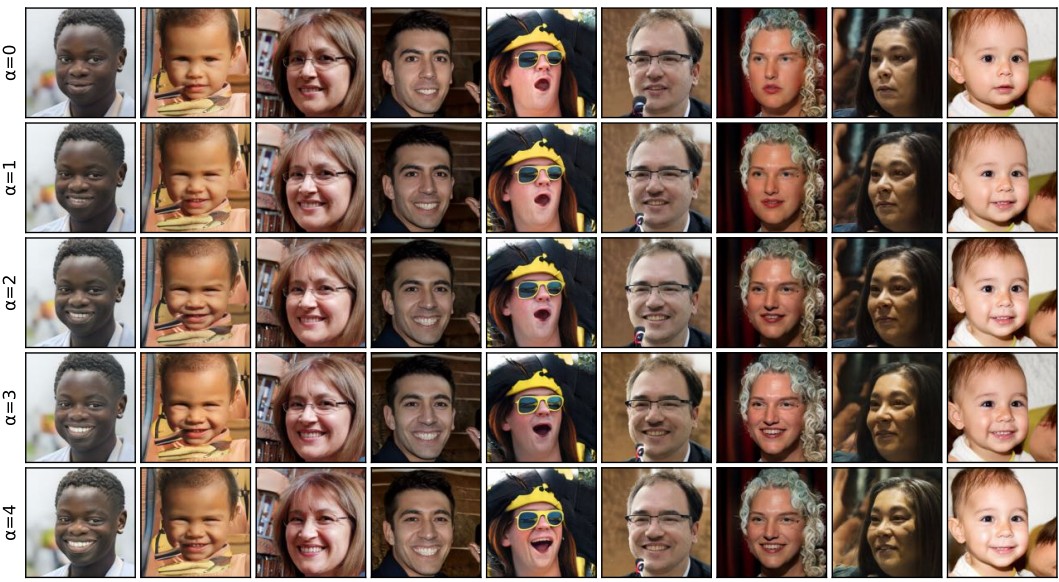

Figure 27: Varying the augmenting power $\alpha$ when making faces happy to show the controllability

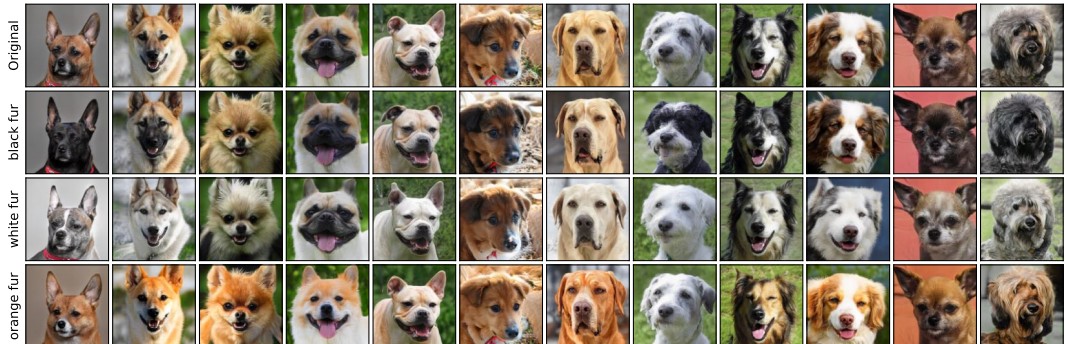

Figure 28: Text-guided physical fur color editing

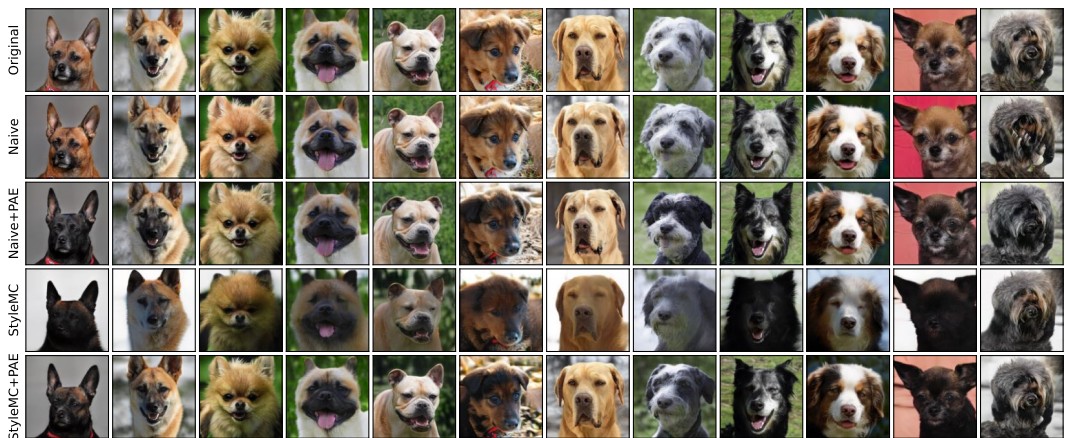

Figure 29: Comparison of different models. The text prompt is "black fur".

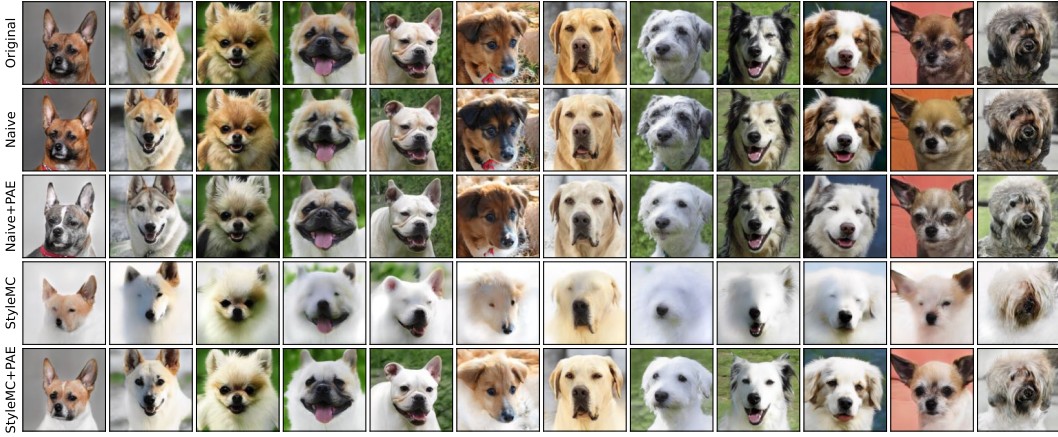

Figure 30: Comparison of different models. The text prompt is "white fur".

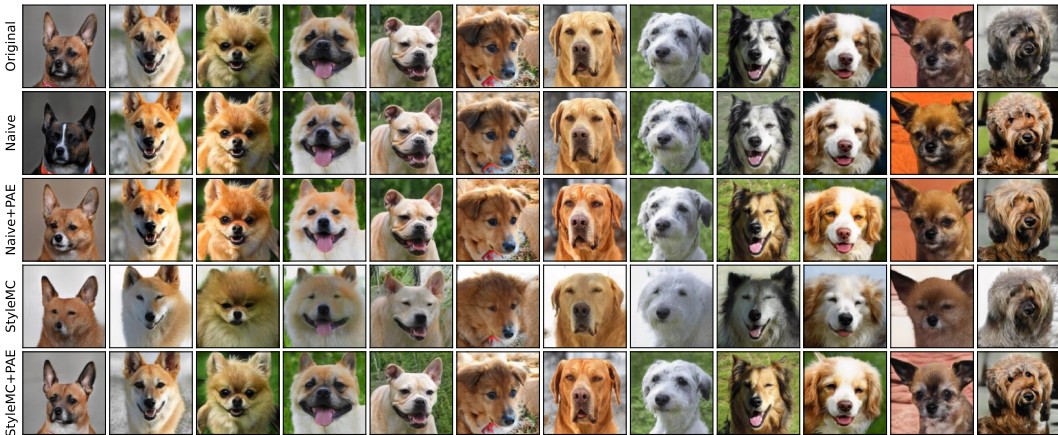

Figure 31: Comparison of different models. The text prompt is "orange fur".

