# OpenReview forum: "CLIP-PAE: Projection-Augmentation Embedding to Extract Relevant Features for a Disentangled, Interpretable and Controllable Text-Guided Image Manipulation"
_ICLR.cc/2023/Conference — Submitted to ICLR 2023_

### Official Review · Reviewer_62cf · 2022-10-24

**Confidence:** 4
**Correctness:** 3
**Technical Novelty And Significance:** 2
**Empirical Novelty And Significance:** Not applicable
**Recommendation:** 5

**Clarity, Quality, Novelty And Reproducibility:**

This paper is well written and easy to follow. The proposed method is reasonable and novel in limited scenarios. Considering the simplicity of the method, the reproducibility should be high.


**Details Of Ethics Concerns:**

As discussed in Sec. 6, the attributes are limited in the training data. In other words, attributes that have not appeared in the dataset cannot be handled. This shows the biasness of the method. Moreover, the methods is limited to only certain scenarios as I showed in the weakness.

**Strength And Weaknesses:**

Strength:
1.	This paper is well writing and easy-to-follow.
2.	The analysis of embedding space well explains the motivation of the approach.
3.	The proposed component (PAE) is reasonable and simple enough for re-implementation. It can be combined in other models.
4.	Experimental results show the effectiveness of the proposed PAE in text-guided face editing.

Weakness:
1.	The experiments are only conducted on face datasets which makes the title “text-guided image manipulation” over claimed.
2.	Besides Gram-Schmidt and dimension reduction techniques for project, clustering is also a possible solution. While the authors have not mentioned about other alternative methods.
3.	The method seems limited in several scenarios:
a)	Non-facial images, such as animals, architecture or other objects.
b)	Texts that are not attributes. It seems that attribute is better to be mapped into subspace while other texts may be difficult.
c)	Attributes with overlaps. There are some attributes that affect multiple parts in the face. For example, “black hair and wearing glasses”, “Asian woman”.
d)	Attributes that are not existed in training data. This also indicates the biasness of this method.


**Summary Of The Paper:**

This paper aims to achieve disentangled, interpretable and controllable text-guided image manipulation with CLIP-based models. The authors propose CLIP projection-augmentation embedding (PAE) to replace the original multimodal embedding as optimizing target. PAE is simple and can be easily to injected into other CLIP-based text-guided image manipulation models. Experimental results on face datasets shows that it can disentangle face attribute and even control the degree of attribute in the generated results.

**Summary Of The Review:**

Although I could recognize the simplicity and effectiveness of the method in single attribute facial editing, the method has limitation in more broader scenarios of text-guided image manipulation.

---

> ### Author Response · Authors · 2022-11-19
> **Response to Reviewer 62cf**
>
> Thank you for the detailed and constructive comments.
>
> **Q1: Experiments on non-facial images**
>
> Please refer to the general response
>
> **Q2: Alternative methods**
>
> Thanks for suggesting clustering for subspace construction. Indeed, there are many possible ways to perform subspace projection and augmentation. In addition to the ones proposed in the main text, we also experimented with two more options for projection and three more for augmentation in Appendix A.2 and reported their performance in Appendices A.3 and A.6. The methods we reported in the main sections have shown better performance and are simple to work with, compared to the alternatives we tested. We also included a summary of the evaluation of alternative methods in Appendix A.3.
>
> **Q3: Application scenarios**
>
> a) Please refer to the general response for non-facial images
>
> b) We agree that there are cases where the prompt does not associate with an obvious attribute. “Donald Trump”, as pointed out by reviewer WWDd, is one such example. We admit this is a limitation of our method and have added a discussion in Section 6.
>
> c) We admit that this is another limitation of our method. Since PAE works by using subspace to capture the information of a specific attribute, it might work less well if there is more than one attribute to disentangle. However, there are two possible approaches to address this: 1) we can treat the input phrase as several standalone prompts and apply PAE multiple times to edit the image for each attribute; 2) we construct a larger subspace using the text corpus from all involved attributes, but this could be less practical.
>
> d) As our method relies on a pre-trained CLIP, we agree that there will be issues if certain prompts are not in the training data. However, this is a shared limitation of all the prior work we compare with, which also rely on the pre-trained CLIP model. Our objective is to improve the performance of such works that are based on CLIP as the training loss.

---

### Official Review · Reviewer_Qecw · 2022-10-25

**Confidence:** 4
**Correctness:** 3
**Technical Novelty And Significance:** 2
**Empirical Novelty And Significance:** Not applicable
**Recommendation:** 5

**Clarity, Quality, Novelty And Reproducibility:**

*Clarity:* The authors present their approach in a detailed manner and the narrative is meaningful. Although some parts of the approach section can be made clearer as indicated in the section above. Some of the design decisions made for the augmentation is unclear.

*Quality:* Although authors demonstrate improvements quantitatively, the qualitative results demonstrated sometimes don't seem to show much change for certain attributes. The quality of writing, however, is informative and engaging.

*Novelty:* The ideas themselves might not be entirely novel since they are leveraging basic ideas from linear algebra, but the insights provided are potentially helpful for textual editing of most latent space based generative models.

*Reproducibility:* The equations, architecture and data are explained clear and aids in reproducibility. The code  is also provided to help reproduce the metrics reported in the manuscript.

**Strength And Weaknesses:**

## Strengths

1.  The authors propose a simple and elegant solution. The proposed approach is a general augmentation strategy that can be applied to a large class of text based editing approaches.

2. The manuscript is well written with attention to detail. All the important components of the approach have been adequately explained

3. Adequate treatment of the related literature has been provided to contextualize the proposed approach in relation to similar methods.

3.  Code is provided to aid in reproducibility.

5. The authors provide important insights regarding the discrete regions of space occupied by the text embedding and image embedding in the joint space. The solution accounts for this gap while influencing image space edits through text.

6. The appendix provides much needed additional details regarding similarity of text and image embeddings and how to structure the projection and augmentation operation.

## Weaknesses

1. There are several simplifying assumptions made regarding (projection+augmentation) operation. Particularly,

&nbsp; &nbsp;&nbsp; &nbsp; a) Equation (2) and (3) assume that the all other attributes are orthogonal to the attributes captured by the basis of the corpus subspace. It is unclear how valid this assumption is in the general case. For instance, extending upon the examples demonstrated in the paper, mouth open->close and the emotion "surprised" may not truly be disentangled.

&nbsp; &nbsp;&nbsp; &nbsp; b) The augmentation operation is akin to a linear operation in the subspace, again relying on the linearity assumption on the part of the space spanned by a particular attribute (say emotion). Is it not more likely that the space of emotion edits forms a nonlinear manifold, such that augmentation (or movement) in that subspace is not necessarily linear? There is some discussion about this in the appendix section, but a more detailed explanation of why the linearity assumption holds would be helpful.

&nbsp; &nbsp;&nbsp; &nbsp; c) The motivation for the augmentation step is unclear. Particularly, if $w$ is a projection of $e_I$ onto a subspace where all other attributes are disentangled why do we need to explicitly "weaken" components of $w$.

&nbsp; &nbsp;&nbsp; &nbsp; d) A clearer explanation of what exactly equation (8) and (9) are doing would be insightful . Particularly, it seems like the $l_1$ norm of the $e_1$ in prompt-space is being shrunk and then a scaled version of $P(e_T)$ is being added back. Explanation regarding the function of this scaling term (or reference to works which employ a similar strategy to move in subspaces) would greatly improve readability of that section.

2. Do we need a different projection operation for every attribute to be controlled?
4. Experiments are demonstrated on a limited dataset. The narrative would greatly benefit from showing more examples on a wider variety of datasets like (CUB/ Flowers), as it would highlight the generalizability of the approach.

&nbsp; &nbsp;&nbsp; &nbsp;	a) Is the limited dataset an issue of annotated data?

&nbsp; &nbsp;&nbsp; &nbsp;  b) Are the limitation due to lack of pre-trained models for other datasets?

5. A number of references to "optimization"  has been made. What exactly is the optimization process involved ? Since all the steps involved in both the projection and augmentation operators are deterministic and the pre-trained models being used are deterministic, what are the learnable/ optimizable parameters in the system.

**Summary Of The Paper:**

The authors present an approach to edit images using text promp by projecting the image embedding to a subspace which is semantically meaningful to the concept being edited. The method integrates seamlessely into most popular CLIP based text driven editing approaches. State of the art quantitative results are shown for text based image editing on a dataset of faces.

**Summary Of The Review:**

The authors provide a simple and elegant method for text based editing by projecting CLIP based embeddings  onto "meaningful" subspaces. However, the work would greatly benefit from better motivation of the projection and augmentation strategy. Furthermore, the results are only provided on a single class of data. Providing on few other object classes would potentially help aid the narrative for generalizability.

---

> ### Author Response · Authors · 2022-11-19
> **Response to Reviewer Qecw Part 1**
>
> Thank you for the detailed and constructive comments.
>
> **Q1: Simplifying assumptions**
>
> a) Yes, there are certainly correlations between attributes. Mouth openness and surprisingness are such examples. Changes in gender and age may also correlate with changes in hairstyle as well. We agree that the PAE only disentangles attributes from other semantically orthogonal attributes, not necessarily correlated ones. However, this is an inherent ambiguity in the problem formulation of disentanglement. For example, if the task is to modify age, it is not clear how associated changes in hairstyle should be considered disentangled or not. It is not our objective to pursue perfect disentanglements in such a scenario but to improve the level of disentanglement of prior work in general scenarios where disentanglements can be concretely defined. In both the empirical analysis (section 3) and comparative study (section 5), we show that PAE does improve the level of disentanglement of all methods. While PAE does not resolve the inherent ambiguity in disentangling “open mouth” from “surprised”, it is important to compare the results with the models that do not have PAE (figures 16 and 25) and see the improvement in disentanglement PAE brings.
>
> b) We assume linearity because 1. it is simpler and more efficient to compute; 2. it works well in our pilot experiments, analytical studies, and real applications in text-guided image manipulation; and 3. there exist some other methods utilizing the linear structure of the CLIP space (e.g.: https://arxiv.org/abs/2108.00946). However, since these results are empirical demonstrations rather than rigorous mathematical proofs, as you pointed out, in section A.2, we also included some other options for projection and augmentation when the space is assumed to be highly nonlinear. Comparisons of their performances were included in section A.3.
>
> c) While the other irrelevant attributes are disentangled in the subspace, the
> subspace still contains relevant attributes that are not intended for the task. For example, if the prompt is “happy”, the subspace is constructed by not only “happy” but also “sad”, “angry”, etc. The motivation of augmentation is to weaken the components of “sad”, “angry”, etc. and strengthen the “happy” component of $w$.
>
> d) As you noticed, equation (9) mainly consists of two parts: the first term weakens all the components of the projected embeddings, while the second term adds back the projected text in a way that the sum of the coefficients is preserved (equation (8) is just calculating the coefficient $d_k$ of the projected text embedding for (9)). Together, these two terms are equivalent to weakening the components that are small in the projected text. We explained in (c) the reasoning behind this. In order to maintain an embedding that does not deviate too much from the original embedding, we add a constraint that the sum of the coefficients should be preserved.
>
> **Q2: Projection operation**
>
> We provide three different options for the projection operation for different groups of corpus/attributes (e.g., emotion, hairstyle), but it is the same projection for each prompt (e.g., happy, sad, angry) under the same group. In our method, the selection of the corpus actually functions as an explicit specification of the intended attributes to be disentangled in a task. For example, when the objective is to disentangle hairstyle change (from lighting, identity, etc.), we construct the subspace with hairstyle corpus (e.g., long hair, blonde, bald, etc.). When the objective is to disentangle emotion, we construct the subspace with an emotion corpus (e.g., happy, sad, surprised, etc.). Compared to prior work where disentanglement can only be achieved on a fixed set of attributes, our method provides a more flexible and explicit manner to specify any semantically meaningful disentanglements. Specifying the corpus for a particular task/attribute is not difficult given the vastly available text resources on the Internet.
>
> **Q3: Evaluation on non-facial images**
>
> Please refer to the general response.

---

> ### Author Response · Authors · 2022-11-19
> **Response to Reviewer Qecw Part 2**
>
> **Q5: Optimization process**
>
> As shown in Figures 4a and 4b, our PAE provides a better way to formulate the optimization loss. Methods such as StyleCLIP, StyleMC, and TediGAN follow the same training procedure as shown in Fig 4a, only differing in their formulation of the learnable parameters. PAE can be integrated into this pipeline with any learnable parameters, which depends on the choice of the underlying model where PAE is applied. For example, in CLIP-based text-guided image manipulation, the Naive approach is to edit the latent code of a certain generative network so that the embedding of the generated image is similar to the embedding of the given text in the CLIP space, then in this case the learnable parameter is the latent code, and so is it in Naive+PAE approach.  As another example, if PAE is applied to StyleMC or StyleCLIP, the learnable parameters are an edit direction for the image latent codes.

---

### Official Review · Reviewer_WWDd · 2022-10-25

**Confidence:** 3
**Correctness:** 3
**Technical Novelty And Significance:** 4
**Empirical Novelty And Significance:** 3
**Recommendation:** 6

**Clarity, Quality, Novelty And Reproducibility:**

The writing is clear and different variations of the proposed method are discussed. Quality and originality are good.

**Strength And Weaknesses:**

Strengths:
1. I really appreciate the analysis in Section3. It contains intuitive visualization and a preliminary pilot study which provide an experimental view of why the original CLIP score doesn't work well in guiding image generation.
2. The methodology, PAE, is novel. It subtly utilizes the relevant semantics in text to build basic vector subspace.
3. The proposed subspace embedding can be generally used in many previous works that employ CLIP similarity for text-guided image manipulation. Also, the experimental results show that adding the proposed embedding can indeed improve the generation quality.

Weaknesses:
1. My concern in terms of methodology is that I still cannot get why the disentanglement is guaranteed in the proposed method, neither from theoretical proof nor intuitive understanding. If you have a look at Fig3 b, you can find the similarity score in the hairstyle subspace somehow also somehow increases. From the methodology, it does build the basic vectors by embeddings in the separate semantics but there might be some spurious/bias between selected semantics and other unwanted semantics, which is not specifically avoided.
2. It's easy to find the relevant corpus of emotion or haircut subspace. But not all subspaces contain a clear definition of the relevant corpus. For example, for `Donald Trump`, as an example in StyleCLIP, it's hard to find a comprehensive and precise relevant corpus.
3. The quality of the corpus matters for the proposed methods. However, it seems the corpus is manually added. There is no clear indicator for a good set of corpus until it's tried in the experiment and takes back the performance. Such a limitation hurts the generalization ability of the proposed method.
4. In experiments:
- From Fig5a, we can find with PAE, the unwanted semantics also change sometimes. For example, the color of the boy's hoodie changed to blue in StyleCLIP+PAE. So echoing what I mentioned before, entanglement seems still exist. It's not guaranteed that the samples are not cherry-picked. I wonder if there is any way we can quantitatively measure the disentanglement degree?
- In Table 1, it's not in which \alpha, the numbers are obtained. For the different metrics of same method, is the same \alpha used?


**Summary Of The Paper:**

This paper proposed CLIP-PAE, which projects and augments CLIP image embedding and text embedding in different subspaces, and uses embedding in subspace to optimize the cosine similarity for text-guided image manipulation. It's applied to different baseline methods and outperforms them in both quantitative and qualitative results.

**Summary Of The Review:**

Although some details about methodology and experiments, and limitations are not illustrated, the novelty, intriguing analysis, and strong performance make the paper above the bar from my point of view.

---

> ### Author Response · Authors · 2022-11-19
> **Response to Reviewer WWDd Part 1**
>
> Thank you for the detailed and constructive comments.
>
> **Q1: Disentanglement**
>
> The subspace is constructed by the span (or an approximation to the span, depending on different projection methods or whether the subspace is assumed to be linear) of the embedding of the text in a corpus. It turns out that this subspace would only preserve information on attributes relevant to the pre-selected corpus and discard information on other attributes, as detailed in the experiment of Section 3.2. In this way, the subspace disentangles the attributes represented by the pre-selected corpus.
>
> We agree that perfect disentanglement is not guaranteed. The subspace projection does not discard the entire information of irrelevant attributes as you pointed out, but our objective is to improve the level of disentanglement when applying PAE to other methods rather than perfect disentanglement. This has been validated in a series of experiments in Section 5. In Figure 3(b), the increase of similarity in hairstyle space is possibly due to the internal variation of the frames from neutral to LOL (e.g., the similarity in CLIP space is also increasing), the non-perfectness of CLIP encoding the image and text message, and admittedly, the non-perfectness of our subspace’s disentangling ability because of e.g., the information noise in approximating the bases of the subspaces.
>
> We also agree that the selected semantics and other unwanted semantics could have a correlation, such as “open mouth” and “surprised”, as pointed out by reviewer Qecw. We did not specifically address this issue but assume that the embeddings of the images/texts in the CLIP space corresponding to “surprised” and “open mouth” are not overlapping. Therefore, the subspace constructed by the selected semantics would still capture different information from other unwanted but correlated semantics.
>
> **Q2-3: Pre-selection of corpus subspaces**
>
> Indeed, not all prompts are associated with a clear definition of the relevant corpus. This is due to the inherent ambiguity in the problem formulation. If the prompt is Donald Trump, it is not clear what attributes are intended to be disentangled — identity change involves changes in hairstyle, age, gender, race, etc. In our method, the selection of the corpus actually functions as an explicit specification of the intended attributes to be disentangled in a task. For example, when the objective is to disentangle hairstyle change (from lighting, identity, etc.), we construct the subspace with hairstyle corpus (e.g., long hair, blonde, bald, etc.). When the objective is to disentangle emotion, we construct the subspace with an emotion corpus (e.g., happy, sad, surprised, etc.). Compared to prior work where disentanglement can only be achieved on a fixed set of attributes, our method provides a more flexible and explicit manner to construct any semantically meaningful disentanglements.
>
> When it comes to this specific example where Donald Trump is used as the prompt, we could potentially use different celebrity names to construct the corpus space. However, we admit that it is not clear what attributes would be captured in this subspace, as with the inherent ambiguity in what attributes should be disentangled in this task.
>
> We agree that there is no clear indicator for a good set of corpus until it's tested inexperiments and evaluated in  performance. However, we did include all the experiments we have conducted to modify four groups of attributes: emotion, hairstyle, eye, and mouth. They all reasonably perform well and improve the original methods. As it is impossible to perform an exhaustive investigation over all the tasks, we agree that an indicator to predict the quality of the selected corpus could be an important future work.

---

> ### Author Response · Authors · 2022-11-19
> **Response to Reviewer WWDd Part 2**
>
>
> **Q4: Experiments**
>
> We did not aim for perfect disentanglement but relative improvements when applying PAE to other methods. We did include a quantitative evaluation in Section 5.3 on all the results to demonstrate this. We also included all the data and results in the supplementary.
>
> Measurements of disentanglement on resulting images are not easy. One practice [1, 2] is to train a set of image classifiers to predict the attributes that are intended to be disentangled. For example, if we assume facial expression, hair style, gender, identity, etc. to be independent factors, we can train a separate image classifier to predict each of these attributes. After the image manipulation, we apply these classifiers to the resulting images. The classification error on the manipulated attribute would reflect the accuracy of our method. The classification error on the non-changed attributes would reflect the quality of disentanglement. We reported the quantitative results on this evaluation in Table 1, shown as Acc-C and Dis-C.
>
> For different metrics of the same method, we used the same alpha value, which is 7.0. Thanks for pointing it out — we have now noted this in the revision.
>
> [1] Choi, Yunjey, et al. "Stargan: Unified generative adversarial networks for multi-domain image-to-image translation." Proceedings of the IEEE conference on computer vision and pattern recognition. 2018.
>
> [2] Gabbay, Aviv, and Yedid Hoshen. "Scaling-up disentanglement for image translation." Proceedings of the IEEE/CVF International Conference on Computer Vision. 2021.

---

### Official Review · Reviewer_DXMN · 2022-10-25

**Confidence:** 4
**Correctness:** 3
**Technical Novelty And Significance:** 2
**Empirical Novelty And Significance:** 2
**Recommendation:** 5

**Clarity, Quality, Novelty And Reproducibility:**

The paper might lack sufficient details about the proposed method, which causes difficulties to reproduce the results.

**Strength And Weaknesses:**

Strength:

1. The paper is well written and easy to follow.
2. The proposed method can be adopted in most CLIP-based methods.


Weaknesses:
1. Authors only evaluate the limitation of CLIP on a face dataset, which is limited and might not be applied for a general dataset.
2. Figure 1 only shows the distance between image features and original text embeddings, and image features and PAE text embeddings, respectively, which might not be enough to support the proposed problem of CLIP, because (1) it is a 2D visualization using PCA, which might not fully show the real distance in a high-dimensional space (2) the distance between them is hard to quantify the differences, without any reference, e.g., comparison with images from another categories.
3. I am confused about why the proposed method allows a better disentanglement, and how to build the subspace? Do authors need to build multiple subspaces for different attributes, e.g., hair and expression?

**Summary Of The Paper:**

The paper explores the limitation of using CLIP to build a joint space for text and image, where the disentanglement, interpretability, and controllability are hard to guarantee in text-guided image manipulation. To mitigate this problem, a PAE is proposed, which defines corpus subspaces spanned by relevant prompts to capture specific image characteristics. Experiments of adopting the proposed PAE in current CLIP-based method improve their performance qualitatively and quantitatively.

**Summary Of The Review:**

See above weaknesses. I am happy to raise my rating based on authors' responses.

---

> ### Author Response · Authors · 2022-11-19
> **Response to Reviewer DXMN**
>
> Thank you for the detailed and constructive comments.
>
> **Q1: Evaluation on non-facial images**
>
> Please refer to the general response.
>
> **Q2: Demonstration of limitations of CLIP**
>
> (1) We included the 2D visualization for a more straightforward demonstration as it is not easy to visualize distances in high dimensions. We did record in Table 2 (copied below) in Appendix A.1 quantitative results of inter-modality and intra-modality cosine similarities for a more strict demonstration, which reveals the same observation that the inter-modality similarity is larger than the intra-modality one regardless of the semantic meaning. We referred to this table in Section 3.1 in our earlier submission but we are happy to consider moving it to the main section if necessary.
>
> |                               | **face text** | **emotion text** | **hair text** | **face image** | **PAE in emotion subspace** | **PAE in hairstyle subspace** |
> |-------------------------------|---------------|------------------|---------------|----------------|-----------------------------|-------------------------------|
> | **face text**                 | 0.852         | 0.862            | 0.779         | 0.200          | 0.684                       | 0.630                         |
> | **emotion text**              |               | 0.877            | 0.784         | 0.198          | 0.693                       | 0.633                         |
> | **hair text**                 |               |                  | 0.764         | 0.201          | 0.634                       | 0.622                         |
> | **face image**                |               |                  |               | 0.567          | 0.491                       | 0.493                         |
> | **PAE in emotion subspace**   |               |                  |               |                | 0.771                       | 0.731                         |
> | **PAE in hairstyle subspace** |               |                  |               |                |                             | 0.725                         |
>
> (2) We have updated the paper to include more comparisons with images from other categories. In particular, we plot the embeddings of 100 random images and 100 random texts visualized using PCA in Figure 6 and also record the similarity in Table 3 (copied below).
> We note the same observation that image embeddings and text embeddings lie in different regions in the CLIP space and have lower inter-modality similarity compared to the intra-modality similarity.
>
> |           | **image** | **text** |
> |-----------|-----------|----------|
> | **image** | 0.644     | 0.199    |
> | **text**  |           | 0.845    |
>
> **Q3: Subspace formulation**
>
> The subspace is constructed by the span (or an approximation to the span, depending on different projection methods or whether the subspace is assumed to be linear) of the embedding of the text in a corpus. It turns out that this subspace would only preserve information on attributes relevant to the pre-selected corpus and discard information on other attributes, as detailed in the experiment of Section 3.2. Therefore, the subspace disentangles the attributes represented by the pre-selected corpus.  We do need to build multiple subspaces to disentangle different attributes, but this is the key --- the explicit selection of the corpus provides explicit control over how the disentanglement is intended. For example, when the objective is to disentangle hairstyle change (from lighting, identity, etc.), we construct the subspace with hairstyle corpus (e.g., long hair, blonde, bald, etc.). When the objective is to disentangle emotion, we construct the subspace with an emotion corpus (e.g., happy, sad, surprised, etc.). Compared to prior work where disentanglement can only be achieved on a fixed set of attributes, our method provides a more flexible and explicit manner to construct any semantically meaningful disentanglements. The method and the procedure are exactly the same for different attributes except for the selection of the corpus.

---

### Author Response · Authors · 2022-11-19
**General Response**

We thank all the reviewers for their detailed and constructive comments. We address some commonly raised issues below.

**Evaluation on non-facial images**

We are happy to soften our claim and update our contribution to "improving text-guided face image manipulation" as we agree that our case study on facial images alone is not sufficient for a general claim.

We chose facial images for our case study as this is one of the most well-studied topics in image-to-image translation with ample datasets, prior work, and evaluation metrics. For example, one practice to quantitatively evaluate disentanglement on resulting images is to train a set of image classifiers to predict the attributes that are intended to be disentangled. For instance, if we assume facial expression, hairstyle, gender, identity, etc. to be independent factors, we can train a separate image classifier to predict each of these attributes. After the image manipulation, one can apply these classifiers to the resulting images. The classification error on the manipulated attributes would reflect the accuracy of the method. The classification error on the non-changed attributes would reflect the quality of disentanglement. Pre-trained classifiers on facial images are easier to find than other types of images. We also applied an identity preservation metric to evaluate whether the identity of the face has been preserved/disentangled in manipulation. These metrics are not readily available in other tasks. In order to perform a thorough assessment of our approach, we chose to focus on facial images in our experiments.

Nevertheless, as our methodology is not constrained by the domain of input images, we performed one more experiment where we modify the fur color of dogs based on text prompts, as a pilot study of generalizing our approach to non-facial images. We have included the new results in appendix A.7.

**Contribution**

The main motivation of this paper is to point out the inherent limitations of using CLIP as a naive optimization loss in a series of works. CLIP has not only been used in text-guided image manipulation but has also been widely used in text-driven 3D manipulations. As it is not trivial to see that the embedding of the texts does not represent the embedding of the true target image and there are increasingly more papers applying the naive CLIP loss in their methods, it is important to raise the issue to the community and seek better solutions. Our PAE approach is the first attempt to resolve this issue. While it has its limitations in achieving perfect disentanglement in certain scenarios, as pointed out by the reviewers and we address below, our objective is to improve the relative level of disentanglements in facial image manipulation for methods that apply CLIP as their training loss naively. Our case study has shown consistent improvements both qualitatively and quantitatively.

---

### Decision · Program_Chairs · 2023-01-20

**Decision:**

Reject

**Justification For Why Not Higher Score:**

N/A

**Justification For Why Not Lower Score:**

N/A

**Metareview: Summary, Strengths And Weaknesses:**

This paper proposed a new approach with CLIP to enforce better disentanglement, interpretability and controllability for text-guided image manipulation. Reviewers have raised major concerns about weak technical contributions (weak novelty and lack of strong theory justification) and weak empirical studies (e.g., the datasets are restricted only to face while the paper is claimed to address general image domains, lack of comprehensive ablation studies to justify why and when it works or not), etc. Authors have tried to address some of review questions in their rebuttal, however, the work may need more a thorough revision and improvement before it can be ready for acceptance.